

# Tree growth and its climate signal along latitudinal and altitudinal gradients: comparison of tree rings between Finland and Tibetan Plateau

Lixin Lyu[1], Susanne Suvanto[2], Pekka Nöjd[2], Helena M. Henttonen[3], Harri Mäkinen[2], Qi-Bin Zhang[1]

[1]State Key Laboratory of Vegetation and Environmental Change, Institute of Botany, Chinese Academy of Sciences, Beijing, 100093, China
[2]Natural Resources Institute Finland, Bio-Based Business and Industry, Jokiniemenkuja 1, 01370 Vantaa, Finland
[3]Natural Resources Institute Finland, Economics and Society, PO Box 2, 00791 Helsinki, Finland

*Correspondence to*: Lixin Lyu (lixinlv@ibcas.ac.cn)

**Abstract.** Latitudinal and altitudinal gradients can be utilized to forecast the impacts of climate changes on forests. To improve the understanding of forest dynamics on these gradients, we tested two hypotheses: (1) the change in the tree growth-climate relationship is similar along both latitudinal and altitudinal gradients, and (2) the time periods during which climate affects growth the most occur later towards higher latitudes and altitudes. We used tree-ring data from a latitudinal gradient in Finland and two altitudinal gradients on the Tibetan Plateau. We analysed the latitudinal and altitudinal growth patterns in tree-rings and investigated the growth-climate relationships of trees by correlating ring-width index chronologies with climate variables calculated with flexible time-windows, using daily-resolution climate data. The high latitude and altitude plots showed higher correlations between the tree-ring chronologies and growing season temperature. However, the effects of winter temperature showed differing patterns for the gradients. The timing of highest correlation with summer temperatures in southern sites was approximately one month ahead of the northern sites in the latitudinal gradient. In one out of the two altitudinal gradients the timing of strongest negative correlation with summer temperatures at low altitude sites was ahead of the treeline sites, possibly due to differences in moisture limitation. Mean values and the standard deviation of tree-ring width was found to increase with increasing mean summer temperatures on both types of gradients. Our results showed similarities of tree growth responses to growing season temperature between latitudinal and altitudinal gradients. However, differences in climate-growth relationships were also found between the gradients, due to differences in other factors, such as moisture conditions. Changes in the timing of the most critical climate variables demonstrated the need to use daily resolution climate data in studies on environmental gradients.

## 1 Introduction

Understanding how tree growth responds to changes in temperature is crucial to accurately predict the future changes in forest dynamics caused by the continuing global warming. As both altitudinal and latitudinal gradients are associated with



consistent temperature differences, they can serve as natural laboratories to infer forest responses to global warming, using the concept of space-for-time substitution (Jump et al., 2009; Stevens, 1992; Blois et al., 2013). The temperature decline of approximately 5 to 6.5 °C per kilometre of elevation on altitudinal gradient corresponds to moving *ca.* 1000 kilometres poleward on a latitudinal gradient (Jump et al., 2009).

Tree growth responses to temperature changes along latitudinal gradients correspond to changes along altitudinal gradients at mountain areas. At the cold end of the gradients, low temperature limits tree growth and finally prevents the growth, reproduction and survival of trees at the treeline (Henttonen et al., 1986; Körner, 1998; Jobbágy and Jackson, 2000; Lyu et al., 2016b). In contrast, at the low latitude and altitude areas, competition and drought are typical factors limiting tree growth and recruitment (Loehle, 1998; Mäkinen et al., 2003; Lv and Zhang, 2012; Di Filippo et al., 2007). For instance, Mäkinen *et*
*al.* (2003) found that the correlation between radial increment of Norway spruce (*Picea abies* (L.) Karst.) and summer temperature changed from positive near the Arctic Circle to negative in Central Europe. Similarly, by using a multi-species data set from the International Tree-Ring Data Bank, Wettstein *et al.* (2011) found that high-latitude ring-width series were more likely to positively correlate with summer temperatures while low-latitude sites commonly showed negative correlations. On altitudinal gradients, similar correlation patterns between ring-width series and summer temperatures have
been reported, for example, by Andreassen *et al.* (2006) for Norway spruce and by Shen *et al.* (2016) for Changbai larch (*Larix olgensis* A. Henry).

In addition to gradually changing relationship between temperature and growth, the timing of radial increment shifts along the latitudinal and altitudinal gradients (Henttonen et al., 2009; Jyske et al., 2014). The period of cambial activity and xylem cell growth gets gradually shorter towards colder areas due to thermal limits in wood formation (Rossi et al., 2007). As the
20 timing of growth changes, also the time-window, in which climate conditions affect growth most changes with increasing latitude (Henttonen *et al.,* 2014).

However, studies on climate-growth relationships are usually based on monthly average values of climate variables (Briffa et al., 2002; Mäkinen et al., 2002; Andreassen et al., 2006). As the time periods influential to tree growth may not correspond to calendar months, information about climatic effects on tree growth may be lost when monthly averages are used (Hordo et
al., 2011; Korpela et al., 2011; Henttonen et al., 2014). In order to better understand the climatic drivers of tree growth, higher resolution climate data and more flexible time frames should be utilized (Vaganov et al., 1999; Henttonen et al., 2014). In studies along temperature gradients, such as altitudinal and latitudinal gradients, this may be even more crucial because the calendar month time-window may suit some parts of the temperature gradient better than others, and therefore the detected gradient patterns may partly be artefacts of the used time-windows.

In this study, we examined annual ring widths and their climatic signals along two types of temperature gradients, one latitudinal gradient in Finland and two altitudinal gradients on the Tibetan Plateau, using daily-resolution climate data. Our starting hypotheses were: (1) the change in the growth-climate relationship is similar along both latitudinal and altitudinal gradients, and (2) the time periods during which climate affects radial growth the most occur later towards higher latitudes



and altitudes, and, therefore, the use of daily climate data instead of monthly averages reveals more detailed climatic signals in tree growth.

## 2 Materials and Methods

### 2.1 The Finnish sites

In Finland, the study sites were selected across a gradient of over 850 km from southern Finland to the northern timberline of Norway spruce in Lapland (Fig. 1). A total of 48 pure Norway spruce stands in national parks and private forests without visible logging activities were sampled (Table 1). All plots were on mineral soil, typical for Norway spruce in the study area. All sampling sites represent rather similar altitudes (115 m in southern Finland up to 410 m in northern Finland). On each plot, 15 dominant trees with no visible damage were cored at breast height (1.3 m). The number of sample trees was lower in

some cases due to limited number of healthy dominant trees available (Table 1). The sampling and measurements of the dataset are described in detail in Mäkinen *et al.* (2000) for sites 1–40 and in Mäkinen *et al.* (2001) for sites 41–48.

### 2.2 The Tibetan Plateau sites

The data from altitudinal gradients on the Tibetan Plateau consisted of two gradients. The first gradient, sampled in summer 2007, consisted of four eastern Himalaya fir (*Abies spectabilis* (D. Don) Spach) plots on the South Central Tibetan Plateau

(SCTP), ranging from the lower (3410 m) to the upper limit (3920 m) of the Himalaya fir forest (Fig. 1, Table 1). The plots were subjectively located to representative spots regarding topography and stand structure at each altitude level. The plots extended 60 meters along the slope and 30 meters along the contour line. All trees at least 10 cm at breast height (1.3 m) were cored. The data was described in detail by Lv and Zhang (2012). To fill the altitude gap between the uppermost and second uppermost plot, one more plot was established at the altitude of 3800 m and 34 mature trees were cored along a 10 m

wide altitudinal belt (Table 1).

The other gradient in a Smith fir forest (*Abies georgei* var. *smithii* Viguie and Gaussen) on the South Eastern (SETP) Tibetan Plateau was sampled in summer 2013. Four plots were sampled at altitudes ranging from 3900 to 4390 m (Fig. 1, Table 1). The two highest plots were located along the treeline (SE4390 and SE4360) and the two other plots located at the medium (SE4140) and lower (SE3900) altitudes of the fir forest (Fig. 1). For the treeline plots, increment cores were taken from trees

within a 30-meter-width-belt along the treeline, and at least 30 trees were cored for each plot. The lengths of the sampled treeline belt were 260 m and 380 m for SE4390 and SE4360, respectively, according to the stand density. For each of the other two altitudinal plots, six groups of Smith firs were randomly sampled. Each group was composed of a central tree and four nearest trees in different compass directions around the central tree.



### 2.3 Tree-ring datasets

Ring widths were visually cross-dated and measured to an accuracy of at least 0.01 mm and cross-dating was verified using COFECHA software (Holmes, 1983). In the Finnish dataset, a total of 820 tree-ring width series from 48 Norway spruce stands were successfully cross-dated. On the Tibetan Plateau dataset, 401 tree-ring width series from nine fir stands (247 trees from five stands in SCTP and 154 trees from four stands in SETP) were cross-dated. Mean segment length (MSL) range from 67 to 262 in the Finnish data set, and from 74 to 195 on the Tibetan Plateau data set (Table 1).

In order to remove age-related trends, the ring-width series were detrended with a spline function with a 50% frequency response cutoff of 30 years using the ARSTAN software (Cook and Peters, 1981). Ring-width indices (RWI) were subsequently calculated as a ratio between the measured and estimated values. Mean site chronologies were calculated for each plot from the RWIs using the robust bi-weight mean (Cook, 1985).

### 2.4 Climate datasets

Daily climate data (mean temperature and precipitation sum) was obtained from the Finnish Meteorological Institute for the Finnish gradient and from the China Meteorological Data Service Centre (http://data.cma.cn/) for the Tibetan Plateau gradients. Weather stations with a long measurement series were used in order to cover as much of the tree-ring chronologies as possible. We used the Sodankylä, Jyväskylä and Heinola weather stations from the northern, central and southern parts of Finland and Nielamu and Linzhi weather stations from the SCTP and the SETP data, respectively (Fig. 2). On the Tibetan Plateau, the temperature data was adjusted based on altitude of the plots, using monthly temperature lapse rates defined for the South Central Tibetan Plateau (used for SCTP) by Kattel *et al.* (2013) and for the South Eastern Tibetan Plateau (SETP) by Kattel *et al.* (2015).

To make the latitudinal and the altitudinal gradients comparable, we analysed the relationships of tree growth and climate variation on a gradient of mean temperature of the warmest month (July). For the Finnish data set, we derived the mean July temperature for each site from the interpolated climate data set at a spatial resolution of $10 \times 10$ km (Venäläinen et al., 2005). For the Tibetan Plateau data set, we calculated the mean July temperature based on the altitude difference and temperature lapse rates for each plot.

### 2.5 Statistical methods

Mean and standard deviation of ring-widths, as well as mean sensitivity (MS) and first-order autocorrelation (AR1) of the increment index series, were calculated for each plot. Mean sensitivity measures the difference in ring-width from one ring to the next and first-order autocorrelation is the correlation between subsequent increment indices (Fritts, 1976).

To further explore the gradient patterns of tree growth along the latitudinal and altitudinal gradients, we conducted hierarchical cluster analyses for all the site chronologies over the common period 1917-1996. The clustering was based on the Euclidean distance metric and the Ward's method using Matlab software (R2014a, The MathWorks, Natick, MA.).





In order to study the relationship between climate variables and the ring-width indices, we calculated the Pearson product-moment correlation coefficients between the plot-wise mean increment chronologies with temperature and precipitation of the current year and the previous year. Instead of using monthly means, mean temperature and precipitation sum were calculated in moving time-windows of 31 days, i.e., by moving the period forward at the resolution of one day. Correlations

were calculated between the RWIs and climate variables for all possible 31-day-windows from the start of previous May to the end of the August of the growth year. The correlation periods were 1938-1997 and 1968-2006 for the Finnish and Tibetan Plateau gradients, respectively.

The critical time periods of temperature and precipitation were identified for each plot based on the highest correlation with the mean RWI chronologies. The magnitude of the correlations and the timing of the most influential time period were

10 compared separately for four seasons (previous year May to August, previous year September to December, growth year January to April, and growth year May to August), so that the central date of time-window used for calculating the climate variables was located within the season in question. The correlation analysis was conducted in statistical software R 3.23 (R Core Team, 2015).

## 3 Results

The means and standard deviations (SD) of stand level ring-width increased towards south in Finland and towards lower elevations on the Tibetan Plateau (Fig. 3). In contrast, the opposite trends for mean sensitivity (MS) were found between the latitudinal and altitudinal gradients. MS increased towards lower altitudes, but decreased towards lower latitudes in Finland. In addition, weak increasing trends were observed for first-order autocorrelation towards the cold ends of both gradients.

Hierarchical cluster analysis showed that stand level RWI chronologies were clustered by latitude or altitude within each

20 gradient (Fig. 4). In the Finnish dataset, RWI chronologies in north Finland formed one cluster and the chronologies in central and southern Finland another. Within the latter, the series were clustered into two groups by latitude (Tammela in southern Finland and Pyhä-Häkki in central Finland). The tree-ring series from the Tibetan Plateau were clustered into two gradient clusters (SCTP and SETP; Fig. 4). Within these clusters, two sub-clusters were formed following the altitudes of plots: higher altitude and lower altitude plots formed their own clusters.

Tree radial growth was positively and significantly correlated with growing season temperatures at high latitudes (northern Finland) and high altitudes (plots 49-50 of SETP and 53-54 of SCTP), whereas, correlations were weaker or statistically not significant on the lower altitude or latitude plots (Figs 5 and 6d). In addition, the RWIs from high latitudes and altitudes were negatively correlated with early summer (May-June) temperatures, but the correlations were weaker or non-significant for the plots at the warmer end of the latitudinal gradient (Figs 5 and 6). On the Tibetan Plateau, these negative correlations were

accompanied with positive correlations with precipitation in the same time-windows (Figs 5g and 5h).



In Finland, negative correlations between the RWIs and late-winter and early-spring temperatures were found on most plots (Figs 5, 6g). These negative correlations showed a latitudinal pattern, with stronger correlations occurring on the northern plots (Fig. 6g). On the Tibetan Plateau, the correlations between the RWIs and winter temperatures were mainly non-significant, although negative correlations with late winter temperatures occurred on one low-elevation plot in the SETP gradient (Figs 5 and 6g).

Previous autumn temperatures had positive correlations with the RWI chronologies both in Finland and on the Tibetan Plateau, without clear patterns in the magnitude of correlations (Figs 5 and 6b). Negative correlations with previous summer temperatures were found for many of the plots on the Tibetan Plateau and in southern Finland, but not for the high latitude plots in Finland (Figs 5 and 6e). On the Tibetan Plateau, the magnitude of the negative correlation between the RWIs and previous summer temperature did not have an altitudinal trend (Fig. 6e).

The timing of the highest correlation varied between the plots in all seasons (Fig. 7). While in some cases the time-windows with the highest correlation did not differ notably (e.g., the Tibetan Plateau plots in Fig. 7b and d), in other cases the time-windows with the highest correlations were spread throughout the whole season (e.g., the Finnish plots in Fig. 7a and b). The Finnish plots seemed to have larger variability in this respect.

The time-windows of the highest correlations between the RWI chronologies and climate variables were mostly found outside of calendar months (Fig. 7). The correlation analysis using monthly climate data resulted in weaker correlation coefficients and fewer significant correlations than using daily climate data (Figs. 6 and 8).

## 4 Discussion

### 4.1 Radial growth variation

Our results confirmed that ring-widths are lower at the cold ends of both latitudinal and altitudinal gradients. Moreover, the tree radial-growth variations (as indicated by SD) also decreased with increasing latitude and altitude. Interestingly, opposite trends were found for mean inter-annual growth variations (MS) along the two types of gradients. Tree growth is generally supposed to be more sensitive to environmental changes towards the harsher end of an environmental gradient (Fritts, 1976). However, the mean sensitivity decreased as altitude increase on the Tibetan Plateau. This pattern was also found in a recent study on the southern Tibetan Plateau (Lyu et al., 2016a). The underlying mechanism accountable for this pattern is largely unknown. We believe that the temperature gradients might be confounded by local factors, such as drought (Liang et al., 2014) and plant-plant interactions (Lyu et al., 2016b; Liang et al., 2016). The local factors could shape diversified habitats with varying limiting conditions other than temperatures.

The results of the hierarchical cluster analysis confirmed the gradient patterns of growth variation along the latitudinal and altitudinal gradients. For instance, the RWI chronology representing the radial growth variation of the treeline site SE4390 was clustered with the site SE4360, which was located in a different summit, rather than the two lower sites of the same



summit. Tree growth at upper and lower altitudes is likely to be under different climate controls (Lv and Zhang, 2012; Shen et al., 2016).

## 4.2 Climate signals of tree radial growth

Our results suggest that the positive correlations between tree radial growth and growing season temperatures are stronger towards the cold end of the gradients. These results are supported by earlier studies on latitudinal and altitudinal gradients (Mäkinen et al., 2003; Andreassen et al., 2006; Shen et al., 2016). The negative correlations with temperatures during May and June also showed a gradient pattern, with significant correlations occurring mainly on the high altitude and latitude plots (Figs 5 and 6h). In the SETP gradient, this negative correlation was accompanied with a positive correlation with precipitation (Fig. 5f), thus indicating a drought limitation of tree growth. This is also supported by previous studies in the area (Lv and Zhang, 2012). However, in the Finnish plots RWI was not positively correlated with precipitation during the time periods of negative correlations with early summer temperatures. This suggests that these negative correlations were not related to lack of moisture in Finland. Indeed, droughts during spring are rare in Finland due to abundant moisture from the melting of snow. Consequently, the similar temperature correlation patterns in the early summer for the latitudinal and altitudinal gradients have resulted from different underlying mechanisms.

Previous autumn and early winter temperatures were found to be significantly and positively linked with tree growth in all gradients, but with relatively weaker strength compared with summer temperatures. This could be related to carbohydrates produced in the autumns being stored and used for growth in the following growing season (Rammig et al., 2015). However, in our results positive correlations were also found for temperature in November to December, when light levels in Finland are low for photosynthesis and trees are likely to be in winter dormancy (Repo 1992, Beuker et al. 1998). The underlying mechanism is still unclear.

Our results demonstrate the effect of late winter temperatures on radial growth in the latitudinal gradient in Finland, but not in the altitudinal gradients. Significant and negative correlations were found across the whole latitudinal gradient, particularly on the northern plots. Previous studies in these high-latitude regions also showed similar results for Norway spruce forests (Miina, 2000; Mäkinen et al., 2003; Andreassen et al., 2006). Reduced growth in years with mild winters could be related to the timing of spring activation. Early activation from dormancy during warm winters may lead to net carbon loss if respiration losses exceed the production in photosynthesis due to low light levels (Skre and Nes, 1996; Linkosalo et al., 2014). Early activation in spring may also increase the risk of frost damage (Cannell and Smith, 1986; Hannerz, 1994). In contrast, on the Tibetan Plateau the correlations between the RWIs and winter temperatures were mainly non-significant. Only one low-altitude plot of the SETP gradient had significant negative correlation between RWI and temperature around March.





### 4.3 The critical time-windows for climate influence on tree growth

Our results support the hypothesis that the timing of the most influential time-window of climate conditions on tree growth changes along latitudinal and altitudinal gradients. In the southernmost sites of the latitudinal gradient, the time-windows with the strongest positive correlations between RWI and growing season temperature (May-August) were about one month
ahead of the northernmost sites, while no obvious trend was detected in the altitudinal gradients. However, strongest negative correlation between RWI and temperature during the May-August period occurred about 15 days earlier at the lowest altitude site in SETP than at the treeline sites, indicating a possible alleviation of moisture limitation with increasing altitude.

The results demonstrate that the use of daily resolution climate data can provide more details in the studying of tree growth
climate-relationships. The timing of tree growth changes on altitudinal and latitudinal gradients (Rossi et al., 2007; Henttonen et al., 2009; Jyske et al., 2014) and, therefore, the timing of the most influential period is also bound to change. The strength of the relationship between tree growth and climate variables could be underestimated due to the failure in identifying the most influential periods outside of the calendar months. For instance, tree growth on the two high altitude plots (SC3920 and SC3800) in SCTP was significantly correlated with temperatures of the current summer (Fig. 6d), but this
association was not detected when using calendar months (Fig. 8d). Usage of daily-resolution climate data is thus recommended in future studies dealing with growth-climate relationships along environmental gradients.

### 4.4 Other factors affecting tree growth

Latitudinal and altitudinal gradients share a similar decreasing trend in temperature, but many other factors, such as precipitation and light availability, are not necessarily consistent between the gradients (Körner, 2007; Jump *et al.*, 2009).
Even though we avoided drought prone locations in the selection of the study sites for both the Finnish and Tibetan datasets, there were still differences in moisture conditions between Finland and the Tibetan Plateau. For instance, the negative correlations with temperature, together with the positive correlations with precipitation during early summer indicate moisture limitation of tree growth in the Tibetan sites.

Masting (*i.e.,* intensive flowering) is another factor affecting the radial growth variation (Selås et al., 2002; Pukkala, 1987).
In masting years, a reduction of tree growth can be attributed to the production of extensive amounts of seeds, because more carbohydrates are allocated to reproduction (Pukkala, 1987; Hacket-Pain et al., 2015). Our results showed negative correlations between the tree growth and preceding summer temperature in all gradients, possibly because of the positive effect of high summer temperature on flowering intensity during the following year (Koenig and Knops, 1998).





## 5 Conclusions

Our results support the hypothesis that the change in the growth-climate relationship is similar along both latitudinal and altitudinal gradients, especially for the effects of growing season temperature on growth. In addition to temperature, other factors, such as moisture availability and masting events, also affect growth variation and, thus, add uncertainties to the

comparison of the temperature gradients. Therefore, a combined analysis incorporating the effects of these gradient-type-related features with temperature trends merits further investigation.

We demonstrated that the use of daily resolution climate data reveals more accurate information about the climate signals in the tree-ring data than monthly data. The critical time-windows for climatic effects on radial growth occurred earlier at lower latitudes and altitudes than at the cold ends of the gradients. Therefore, the use of daily climate data may disclose gradient

patterns that could not be detected if monthly climate data is used.

## 6 Acknowledgements

This study was supported by the National Natural Science Foundation of China (No. 31361130339, 31330015 and 31300409) and by grants from the Academy of Finland (No. 257641 and 265504). The climate data were obtained from the China Meteorological Data Service Centre and the Finnish Meteorological Institute. All the data resulting from this study are

available from the authors upon request (qbzhang@ibcas.ac.cn for Tibetan subset; harri.makinen@luke.fi for Finnish subset).

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





**Table 1**. Information on the plots, tree-ring samples and the climate conditions along the latitudinal gradient in Finland and the altitudinal gradients on the southern Tibetan Plateau.

| Site | Plot No. | Region code[*] | Latitude (N) | Longitude (E) | Altitude (m) | No. of trees | No. of cores | MSL[#] | MAT[§] (°C) | MAP[§] (mm) | T[July][§] (°C) |
|------|----------|------------|--------------|---------------|--------------|--------------|--------------|-----|------------|------------|-----------|
| **Finland** | | | | | | | | | | | |
| Vuotso | 1 | NFIN | 68° 13′ | 27° 11′ | 300 | 9 | 18 | 163 | -1.77 | 483 | 11.1 |
| Pokka | 2 | NFIN | 68° 08′ | 25° 43′ | 285 | 10 | 20 | 146 | -1.65 | 484 | 11.2 |
| Pokka | 3 | NFIN | 68° 03′ | 25° 39′ | 295 | 11 | 21 | 152 | -0.35 | 543 | 11.2 |
| Pallastunturi | 4 | NFIN | 68° 02′ | 24° 04′ | 410 | 11 | 11 | 211 | -0.94 | 519 | 11.0 |
| Pallastunturi | 5 | NFIN | 68° 00′ | 24° 08′ | 370 | 11 | 11 | 180 | -0.7 | 519 | 11.2 |
| Kittilä, Tieva | 6 | NFIN | 68° 00′ | 25° 43′ | 275 | 10 | 20 | 138 | -0.38 | 543 | 11.4 |
| Vuotso | 7 | NFIN | 68° 00′ | 26° 55′ | 305 | 11 | 22 | 189 | -1.89 | 491 | 11.3 |
| Pallasjärvi | 8 | NFIN | 67° 59′ | 24° 14′ | 330 | 11 | 11 | 67 | -1.74 | 486 | 11.3 |
| Vuotso | 9 | NFIN | 67° 59′ | 26° 35′ | 270 | 10 | 19 | 148 | -1.5 | 488 | 11.5 |
| Pallas | 10 | NFIN | 67° 58′ | 24° 05′ | 405 | 8 | 8 | 173 | -1.67 | 492 | 11.1 |
| Kittilä, Kiistala | 11 | NFIN | 67° 58′ | 25° 39′ | 305 | 10 | 20 | 124 | -1.09 | 500 | 11.3 |
| Soukkavaara | 12 | NFIN | 67° 51′ | 24° 51′ | 305 | 10 | 10 | 191 | -0.36 | 543 | 11.6 |
| Jerisjärvi | 13 | NFIN | 67° 50′ | 23° 59′ | 335 | 9 | 9 | 153 | -1.57 | 487 | 11.5 |
| Pomokaira | 14 | NFIN | 67° 50′ | 26° 25′ | 265 | 11 | 22 | 183 | -1.9 | 495 | 11.4 |
| Kittilä, Saattopora | 15 | NFIN | 67° 47′ | 24° 21′ | 270 | 10 | 19 | 177 | -1.78 | 493 | 11.7 |
| Kittilä, Kumputunturi | 16 | NFIN | 67° 38′ | 25° 32′ | 205 | 6 | 12 | 204 | -1.17 | 499 | 12.0 |
| Sodankylä, Mosku | 17 | NFIN | 67° 37′ | 27° 11′ | 210 | 10 | 20 | 131 | -1.38 | 494 | 11.8 |
| Kittilä, Tepsa | 18 | NFIN | 67° 35′ | 25° 32′ | 202 | 10 | 19 | 167 | -0.9 | 521 | 12.1 |
| Ristonmännikkö | 19 | NFIN | 67° 10′ | 26° 19′ | 245 | 10 | 20 | 209 | -1.9 | 493 | 12.1 |
| Niesi, Karhukuru | 20 | NFIN | 66° 59′ | 25° 54′ | 215 | 13 | 13 | 187 | -1.87 | 480 | 12.4 |
| Niesi, Kunetti | 21 | NFIN | 66° 59′ | 25° 53′ | 270 | 13 | 13 | 258 | -0.9 | 521 | 12.2 |
| Niesi, Kutuselkä | 22 | NFIN | 66° 55′ | 25° 52′ | 255 | 11 | 11 | 262 | 0.04 | 539 | 12.3 |



| | | | | | | | | | | | | | |
|---|---|---|---|---|---|---|---|---|---|---|---|---|---|
| Niesi, Kutuselkä | 23 | NFIN | 66° | 55′ | 25° | 56′ | 275 | 13 | 13 | 170 | -2.14 | 481 | 12.3 |
| Niesi, Turhapuro | 24 | NFIN | 66° | 55′ | 25° | 56′ | 275 | 10 | 20 | 149 | -1.89 | 482 | 12.3 |
| Kivalo | 25 | NFIN | 66° | 19′ | 26° | 40′ | 285 | 11 | 11 | 241 | -2.07 | 478 | 12.5 |
| Kivalo | 26 | NFIN | 66° | 19′ | 26° | 42′ | 240 | 10 | 20 | 202 | -2.27 | 477 | 12.7 |
| Kivalo | 27 | NFIN | 66° | 18′ | 25° | 42′ | 240 | 10 | 20 | 190 | -0.56 | 543 | 12.7 |
| Kivalo | 28 | NFIN | 66° | 18′ | 25° | 42′ | 250 | 9 | 9 | 169 | -0.39 | 543 | 12.6 |
| Kivalo | 29 | NFIN | 66° | 18′ | 25° | 42′ | 245 | 12 | 12 | 192 | -0.81 | 521 | 12.7 |
| Lamu, Mäsäjärvi | 30 | NFIN | 66° | 18′ | 25° | 31′ | 160 | 13 | 13 | 157 | -1.06 | 511 | 13.1 |
| Kivalo | 31 | NFIN | 66° | 18′ | 26° | 45′ | 315 | 10 | 20 | 173 | -0.65 | 544 | 12.4 |
| Kivalo | 32 | NFIN | 66° | 18′ | 26° | 43′ | 310 | 10 | 20 | 199 | -0.64 | 544 | 12.4 |
| Pyhä-Häkki | 33 | SFIN | 62° | 50′ | 25° | 29′ | 175 | 10 | 20 | 185 | 2.4 | 630 | 14.2 |
| Pyhä-Häkki | 34 | SFIN | 62° | 49′ | 25° | 29′ | 162 | 10 | 20 | 86 | 2.4 | 630 | 14.2 |
| Pyhä-Häkki | 35 | SFIN | 62° | 49′ | 25° | 29′ | 162 | 10 | 20 | 124 | 2.4 | 630 | 14.2 |
| Pyhä-Häkki | 36 | SFIN | 62° | 49′ | 25° | 29′ | 166 | 10 | 20 | 96 | 2.4 | 630 | 14.2 |
| Pyhä-Häkki | 37 | SFIN | 62° | 49′ | 25° | 29′ | 166 | 10 | 20 | 134 | 2.4 | 630 | 14.2 |
| Pyhä-Häkki | 38 | SFIN | 62° | 49′ | 25° | 29′ | 162 | 9 | 18 | 126 | 2.4 | 630 | 14.2 |
| Pyhä-Häkki | 39 | SFIN | 62° | 49′ | 25° | 29′ | 172 | 10 | 20 | 120 | 2.4 | 630 | 14.2 |
| Pyhä-Häkki | 40 | SFIN | 62° | 49′ | 25° | 29′ | 172 | 9 | 18 | 92 | 2.4 | 630 | 14.2 |
| Tammela | 41 | SFIN | 60° | 44′ | 23° | 43′ | 110 | 10 | 20 | 82 | 4.29 | 635 | 15.1 |
| Tammela | 42 | SFIN | 60° | 43′ | 23° | 41′ | 110 | 10 | 20 | 80 | 4.29 | 636 | 15.1 |
| Tammela | 43 | SFIN | 60° | 41′ | 23° | 50′ | 120 | 10 | 20 | 126 | 4.26 | 640 | 15.1 |
| Tammela | 44 | SFIN | 60° | 41′ | 23° | 49′ | 120 | 10 | 20 | 122 | 4.26 | 640 | 15.1 |
| Tammela | 45 | SFIN | 60° | 41′ | 23° | 50′ | 120 | 10 | 20 | 108 | 4.27 | 641 | 15.1 |
| Tammela | 46 | SFIN | 60° | 40′ | 23° | 52′ | 115 | 10 | 20 | 172 | 4.3 | 642 | 15.1 |
| Tammela | 47 | SFIN | 60° | 39′ | 23° | 52′ | 115 | 10 | 20 | 147 | 4.31 | 643 | 15.1 |
| Tammela | 48 | SFIN | 60° | 39′ | 23° | 53′ | 115 | 10 | 20 | 171 | 4.31 | 643 | 15.1 |
| **Southern Tibetan Plateau** | | | | | | | | | | | | | |
| SE4390 | 49 | SETP | 29° | 39′ | 94° | 43′ | 4390 | 52 | 52 | 146 | -3.09 | 786 | 6.4 |
| SE4360 | 50 | SETP | 29° | 36′ | 94° | 36′ | 4360 | 41 | 41 | 153 | -2.83 | 786 | 6.6 |



| SE4140 | 51 | SETP | 29° 39′ | 94° 43′ | 4140 | 31 | 31 | 195 | -0.97 | 786 | 8.0 |
| SE3900 | 52 | SETP | 29° 39′ | 94° 43′ | 3900 | 30 | 30 | 160 | 1.06 | 786 | 9.5 |
| SC3920 | 53 | SCTP | 27°50′ | 87°28′ | 3920 | 50 | 50 | 119 | 3.05 | 1113 | 12.0 |
| SC3800 | 54 | SCTP | 27°50′ | 87°28′ | 3800 | 32 | 32 | 141 | 3.67 | 1113 | 12.6 |
| SC3700 | 55 | SCTP | 27°50′ | 87°28′ | 3700 | 45 | 45 | 109 | 4.18 | 1113 | 13.1 |
| SC3520 | 56 | SCTP | 27°50′ | 87°28′ | 3520 | 83 | 83 | 93 | 5.11 | 1113 | 14.0 |
| SC3410 | 57 | SCTP | 27°50′ | 87°27′ | 3410 | 37 | 37 | 74 | 5.67 | 1113 | 14.6 |

* NFIN northern Finland; SFIN southern Finland; SETP Southeastern Tibetan Plateau; SCTP south-central Tibetan Plateau.

# MSL mean segment length (years) of tree-ring samples. It indicates the average breast height age of trees for each plot.

§ MAT mean annual temperature, MAP mean annual precipitation, $T_{July}$ mean July temperature. These three statistics were calculated over the period 1966-1995 and 1971-2000 for the Finnish and the Tibetan Plateau data sets, respectively.





**Figure legends**

**Figure 1:** Location map of the sample plots and weather stations in Finland and on the Tibetan Plateau.

**Figure 2:** Monthly mean temperature and precipitation sum at the weather stations used in the study: Heinola (a), Jyväskylä (b) and Sodankylä (c) in Finland, and Linzhi (d) and Nielamu (e) on the Tibetan Plateau. The shaded area of the marked lines and the error bars of the columns are the 1 SD (standard deviation) of the monthly mean temperatures and the monthly precipitation sum over the recording period, respectively.

**Figure 3:** Stand level statistics of raw tree-ring widths along the latitudinal gradient in Finland (FI) and the altitudinal gradients on the Tibetan Plateau (TP). The lines are linear regression lines. The black colour marks the data for the latitudinal gradient, and the red denotes the data for the altitudinal gradients.

**Figure 4:** The cluster analysis of ring width chronologies in Finland and on the Tibetan Plateau. The plot numbers are consistent with Table 1. The plots 1–48 are from the latitudinal gradient in Finland and were grouped in to two clusters corresponding northern (blue colour) and southern (red colour) Finland; the plots 49–57 are from the altitudinal gradients on the Tibetan Plateau (black colour).

**Figure 5:** Correlation coefficients between the ring-width index chronologies (separate sub-figures for each region, and separate colours for each plot) and mean temperature and precipitation sum in moving time-windows of 31 days. The X-axis is the central day of the time-window used in calculating the climate variables, with the first day of each month marked by ticks. The plot numbers are consistent with Table 1. Note that the colour gradient legend for subpanels a-b is by 2 sites.

**Figure 6:** The magnitude of the maximum (mostly positive) (a-d, upper panel) and minimum (mostly negative) (e-h, lower panel) correlations between the RWIs and temperature against the mean July temperature of each plot. The correlations are shown separately in four seasons, so that the central dates of time-window used for calculating the climate variables is located in (1) previous year May to August (1$^{st}$ column), (2) previous year September to December (2$^{nd}$ column), (3) growth year January to April (3$^{rd}$ column) and (4) growth year May to August (4$^{th}$ column). The black X's denotes the plots in Finland; the red circles and triangles denote the plots from the altitudinal gradients on the south-central Tibetan Plateau (SCTP) and southeastern Tibetan Plateau (SETP), respectively. The grey colour indicates correlations not significant at the 95% level.

**Figure 7:** The central dates of the 31-day time-windows with maximum (mostly positive) (a-d) and minimum (mostly negative) (e-h) correlations between the RWIs and temperature in Fig. 6. The black X mark denotes the plots in Finland and the plots on the south-central Tibetan Plateau (SCTP) are marked with red circles and on the southeastern Tibetan Plateau (SETP) with red triangles. The grey colour indicates correlations not significant at the 95% level.

**Figure 8:** The magnitude of the maximum (mostly positive) (a-d, upper panel) and minimum (mostly negative) (e-h, lower panel) correlations between the tree-ring width indices and monthly mean temperatures in different seasons (columns)



against the mean July temperature of each plot. The black X mark denotes the plots in Finland and the plots on the south-central Tibetan Plateau (SCTP) are marked with red circles and on the southeastern Tibetan Plateau (SETP) with red triangles. The grey colour indicates correlations not significant at the 95% level.





**Figure 1**

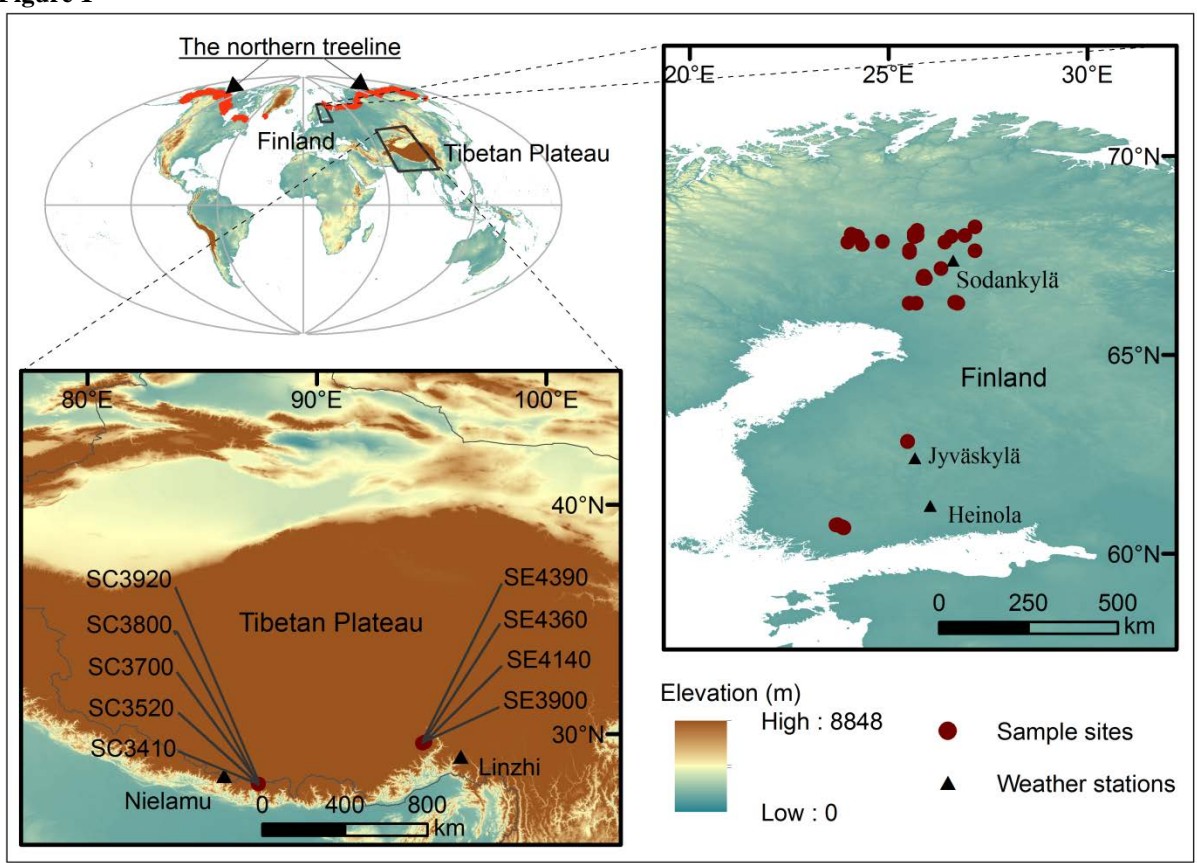





**Figure 2**

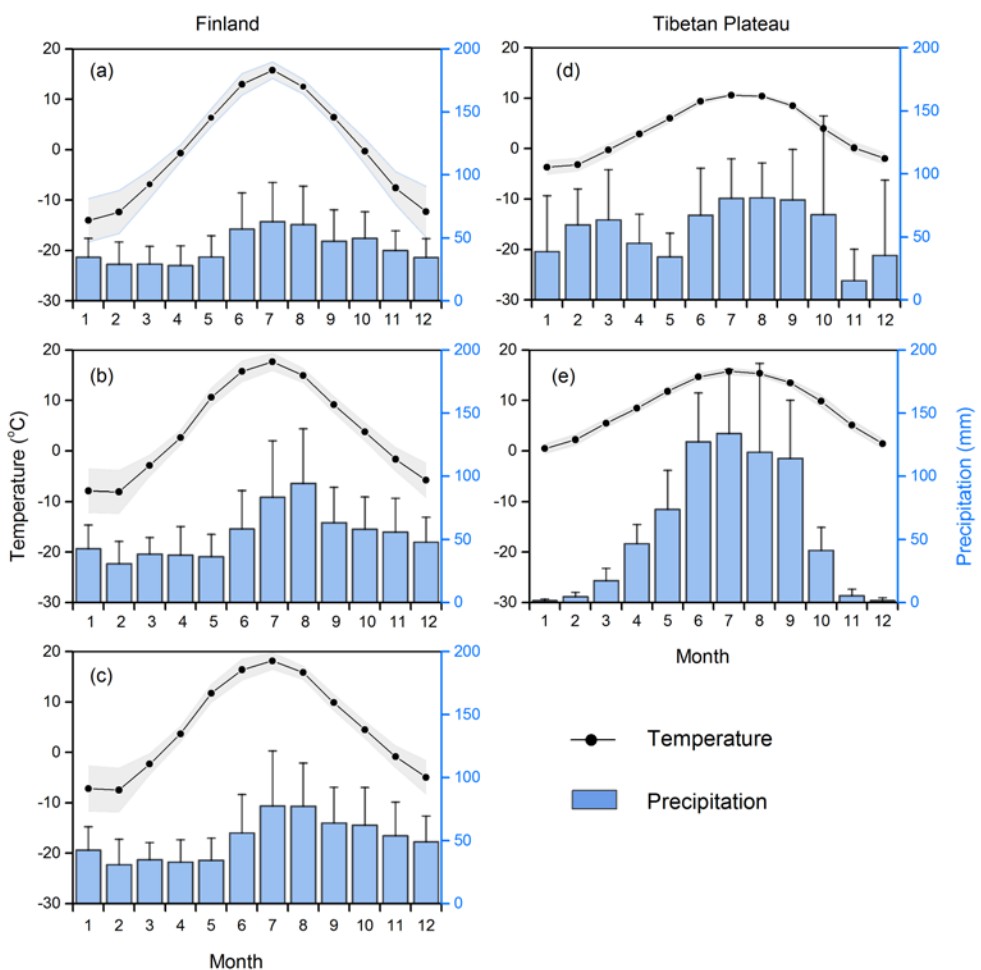



**Figure 3**

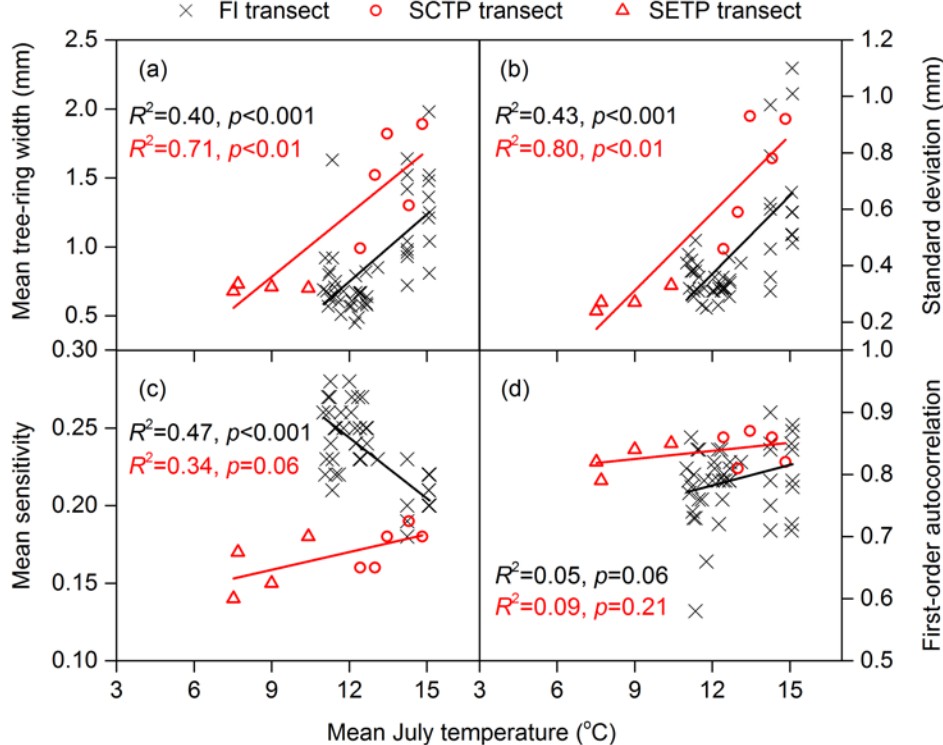





**Figure 4**

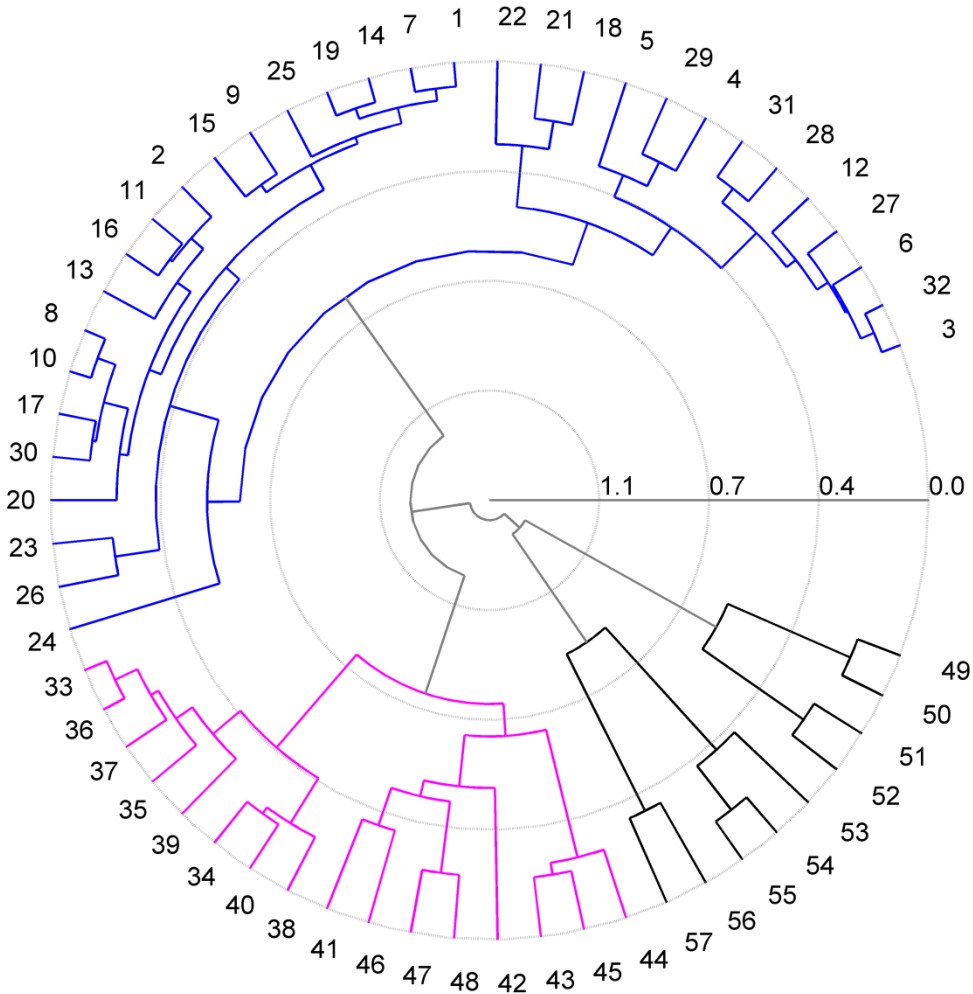





**Figure 5**





**Figure 6**

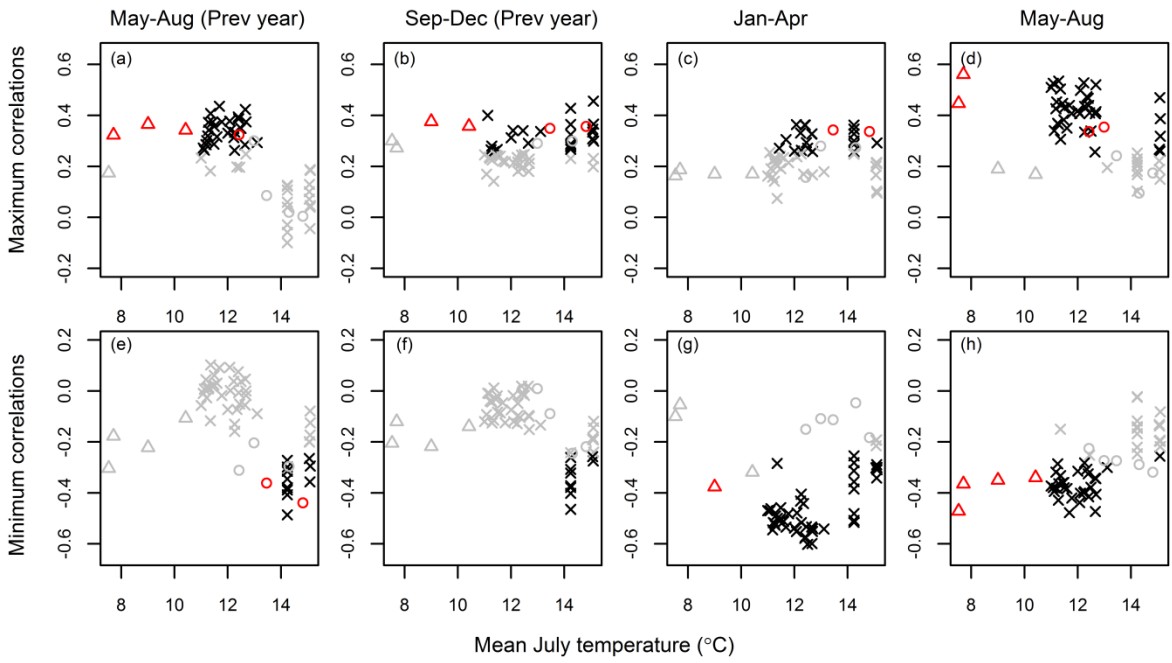





**Figure 7**

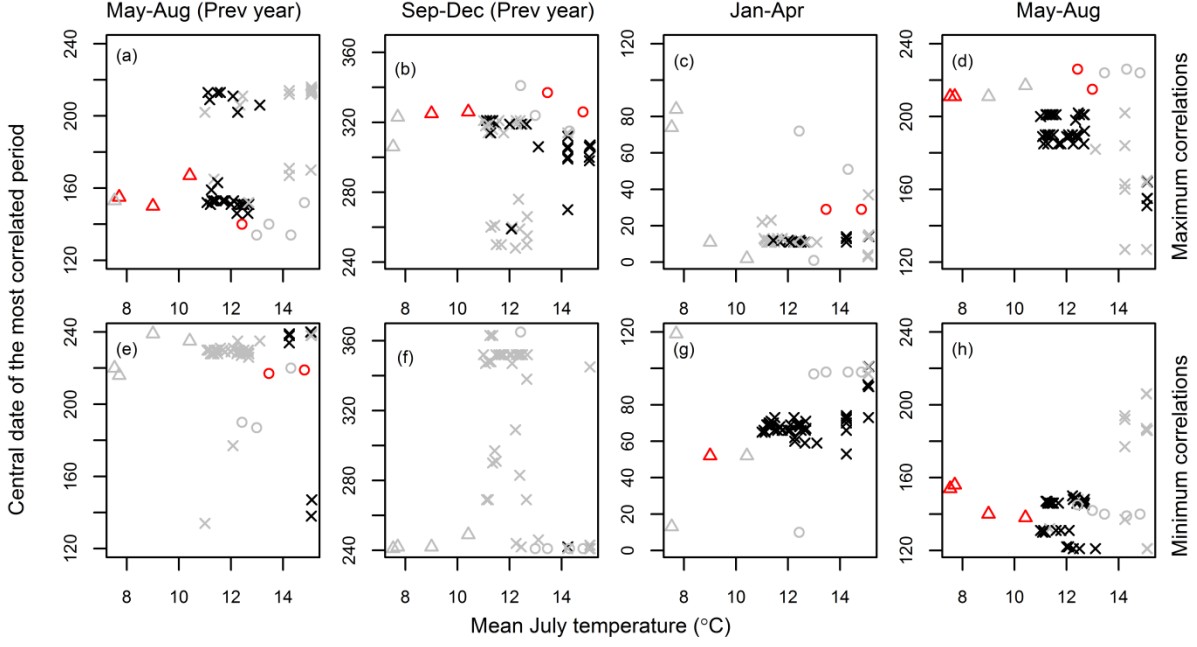





**Figure 8**

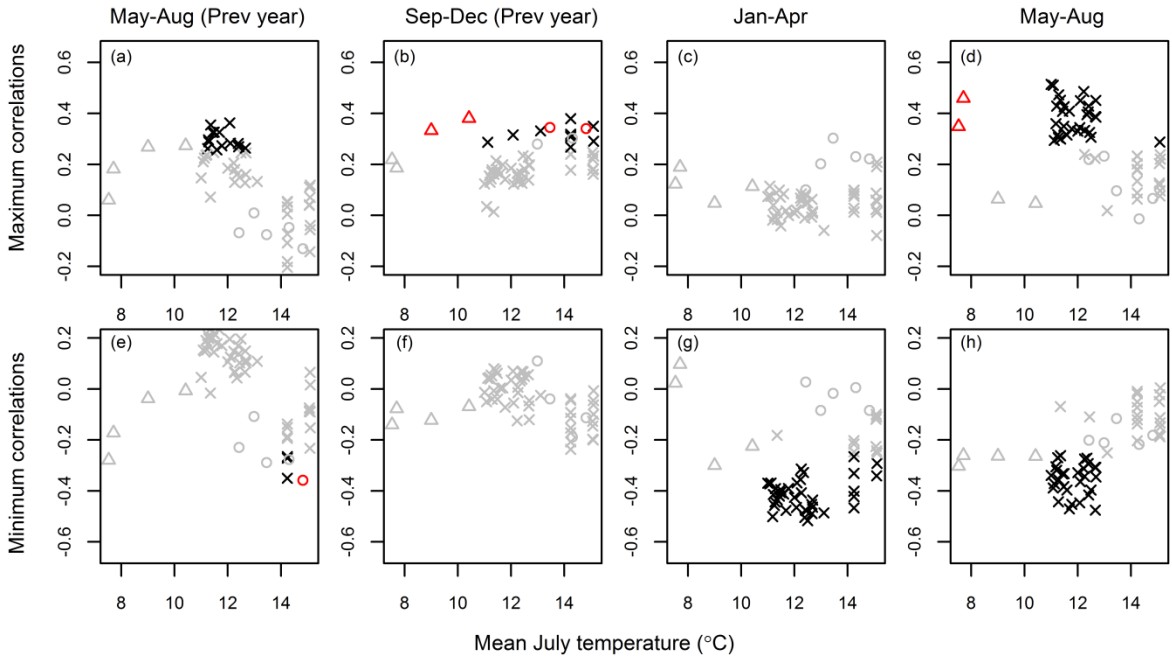

Mean July temperature (°C)