# Peer review of "Tree growth and its climate signal along latitudinal and altitudinal gradients: comparison of tree rings between Finland and Tibetan Plateau"

_Biogeosciences, 2016_

## Referee Comment (RC1) · Anonymous Referee #1 · 17 Feb 2017

General comments: In this manuscript, Lyu et al. showed that tree ring width has a pattern in both the growth (width) and its response to climate (majorly temperature) along with the latitude/altitude gradient. The creative way of using daily-step climate data implies the potential improvement in analyzing such type of time series data, e.g. tree ring analysis. It is interesting to compare/combine the latitude with altitude gradient, and would be very useful in understanding/predicting the change of tree growth in the future "warming" scenario. However, I think more efforts are needed in structuring and surmising of the paper. And I found a bit difficult to follow the results part.

Specific comments:

1 In terms of the climate data used in this paper, "local" meteorological station data

was used in this paper, where temperature was adjusted using lapse rates in altitude gradients. Is this adjustment linearly correction? If so, this correction should have little effect on the correlation analysis. So what's the point to do this adjustment, if no absolute value of the temperature, e.g. GDD, is considered.

2 Multiple climate factors control on tree growth was mentioned in the ms. And a number of (different) temperature signals were explained from the effect of precipitation or soil moisture, which is either non-altitude-corrected or not analyzed. Is it possible to add some analysis or information about this? In the current ms, the mentioned potential drought limitation need more evidence to support.

3 Both the climate signal and the timing for the max signal are changing along with the two gradients. And in the ms, latitude and altitude gradients are both using (July) temperature to distinguish, which implies the changing temperature is one of the major reason for the change of these pattern. If that's the case, the increase of temperature during the record period should also have influence. Does this mean either the timing or the strength of the temperature signal would change with time? Would the warming have any effect on this analysis? Would this weaken the application of using daily climate data? Because it could be always changing year-by-year.

4 There are lots of places mentioned "growing season", e.g. in the abstract, page 5 line 25, page line4 and line17, page 9 line3. However, there is no clear definition about it. If we treat the growing season as May-Aug, which was mentioned in page8 line4, it has both very strong negative and positive correlation between ring width and temperature. However, only positive correlation was mentioned in the ms. Definition and what value was used need to be cleared.

5 Results, especially in page 5, are difficult to follow. It is hard to get consistent described results via reading the figures (Fig. 5 and 6). And the climate signals of "growing season"/"summer" made it more confusing. Does this mean a period of several months, or only one month/peak?

**BGD**

Other comments:

1 Page 3 line 27: Why the special sampling method was applied in these altitude gradient sites.

2 Page 4 line 16: the altitude of these meteorological station should be supplied, especially for the altitude gradient sites. This would be helpful for understanding the process of altitude adjustment, and could be helpful to evaluate the consistence between the local precipitation and met record.

3 Page 5 line 2: what is the difference between plot-wise mean increment chronologies and RWI.

4 Not sure whether Fig. 4 is necessary. The results show more location/site dependent, which is more obvious that these gradients. And this makes sense, because RWI of similar locations should have similar pattern. And in fact, no composite cluster result is used in the following analysis.

5 More caption is needed to describe what is the gray horizontal line in fig.5. If this is the significance test level, why all of the site has the same test level, considering the different length of climate record for different sites.

---

## Referee Comment (RC2) · Anonymous Referee #2 · 10 Apr 2017

The manuscript, "Tree growth and its climate signal along latitudinal and altitudinal gradients: comparison of tree rings between Finland and Tibetan Plateau" has good potential to reveal some insightful ways in which trees and tree populations respond to climate over gradients. I enjoy the potential in the data and some of the findings. It is written well enough for most audiences; some places could use some clarification. While many of the results generally follow prior work, the examination of the sub-monthly response is particularly novel, especially at such large scales.

One major concern I have is how do the authors control for differences in sunlight or day length in comparing to two regions? There are significant differences between the two and can affect the results. How do they ensure that some of the differences they

see, perhaps with the response to precipitation, is not related to day length?

The second smaller concern is that because the authors go to sub-monthly climate responses, Figures 5 or 6 (and much of the discussion related to those figures) are almost unnecessary. The findings are not too unexpected, but the sub-monthly analysis gives more insight. If choosing one, Figure 5 might be preferable over Figure 6. It is easier to read and provides more information.

Suggestion: as this is an international paper, perhaps use only Latin names throughout the manuscript.

There are some concerns with the current status of the work. They are detailed below.

Page 2, lines 1-2: it is likely better to emphasize these are potential natural laboratories. Space for time doesn't really equal time, especially given that less than 100 years are analyzed here. Environmental variability increases with greater periods of time. So removing that from the introduction would be ok. Along these lines, the authors do not fully come back to that concept.

Page 2, Line 7: perhaps "or" instead of "and" between reproduction and survival. Also, here and throughout: the Oxford comma will aid clarity in the manuscript.

Page 2, Lines 8-9: There are updates to the Loehle reference in the region with greater replication at a larger spatial scale. It also indicates the same concept in this sentence. Related: explain here how a negative correlation to temperature and a positive response to precipitation together equals drought.

Page 2, Lines 26-27: (Kim and Siccama 1986) is a great forerunner of this idea and deserves recognition. It was well ahead of its time.

Materials and Methods:

Page 3, Lines16-28: how might sampling in belts alter the climatic response? Does differing densities impact the observed climatic response? See, for instance, (Sánchez-

Salguero et al. 2013, Sohn et al. 2013, Aldea et al. 2017).

Page 4, Line 4 etcetera: the proper convention is crossdate, crossdating, crossdated. Please use this convention.

Page 4, Lines 20-24: it is not likely a serious issue, but how might the inferred temperature with elevation impact the results?

Page 4, Lines 26-29: although long in use, mean sensitivity is not a useful for comparing tree-ring records. See (Bunn et al. 2013). Please remove the MS analysis and comparison from the manuscript.

Suggestion: an interesting comparison might be the coherence within each population over space. Because many of these samples were collected in plots or were aimed to be representative of the forest, perhaps a box plot or something similar expressing the strength of interseries correlation would be compelling and insightful instead of MS.

Page 5, Lines 4-6: why 31 day windows? Did the authors experiment with narrower windows?

Page 5, Lines 8-11: given the submonthly work, it is not clear the use or need for these analyses. Why focus on seasons?

Page 5, Lines 16-19: suggest removing the MS results.

Page 5, Lines 20-24, Figure 4: Did the authors conduct cluster analysis within each region? Might the analysis on all populations force artificial grouping within each region? The analyses in Figures 7 & 8 somewhat supersede the scale of analysis in Figure 4, correct?

Suggestion for Figure 7 and most figures: consider choosing a consistent color scheme for Finland and the Tibetan Plateau and use it throughout the paper. In Figure 4 the TP is black, but by Figure 6, Finland is black. Maintaining the same colors for regions will make it easier on the reader.

Page 5, Lines 25-30, Figure 5: the authors write about growing season, but their information here is more specific. Suggest that in the Results the authors should be more specific. The authors make much of a negative correlation to temperature in early summer, but it appears this response is much stronger in February and March in northern Finland? Am I interpreting this incorrectly? If so, I apologize. If not, consider re-emphasizing these results. They do not seem as similar as suggested in the text. There appears to be negative correlations in northern Finland in May & June, but they are much weaker compared to earlier in the year.

To make Figure 5 easier to interpret, suggest putting the months on the top of the top 2 plots. Also, perhaps make the symbols in Figure 5 smaller or replace with lines so a clearer interpretation can be made.

Page 6, Lines 10-14: why might the Finland plots have larger variability?

Discussion:

Page 6, Line 20: "ring-widths are lower" than what?

Page 6, Lines 21-23: remove MS discussion per above

Page 6, Lines 25-29: samples were collected in plots. Density, diversity, and their impact could presumably be investigated here instead of suggesting they might be at work in the results.

Page 7, Lines 15-19: there is a growing and now somewhat large body work finding or examining the relation between winter temperatures and tree growth. A review of this work would help contextualize the findings here. It might help signify the importance or the continuing line of evidence created by the findings in this study.

Page 8, Lines 9-16: this is one of the most novel aspects of the study here and should be emphasized earlier and more prominently. Interesting findings.

Page 8, Lines 21-23: why might this be? Is there literature that could help account for

this finding?

Page 8, Lines 24-28: Masting comes out from nowhere in this manuscript. Do the species study mast? If so, how regularly? If there cannot be a better tie between the species studied and masting, it is suggested that this section be dropped.

References: Aldea, J., F. Bravo, A. Bravo-Oviedo, R. Ruiz-Peinado, F. Rodríguez, and M. del Río. 2017. Thinning enhances the species-specific radial increment response to drought in Mediterranean pine-oak stands. Agricultural and Forest Meteorology 237:371-383.

Bunn, A. G., E. Jansma, M. Korpela, R. D. Westfall, and J. Baldwin. 2013. Using simulations and data to evaluate mean sensitivity ($\zeta$) as a useful statistic in dendrochronology. Dendrochronologia 31:250-254.

Kim, E., and T. G. Siccama. 1986. The influence of temperature and soil moisture on the radial growth of northern hardwood tree species at Hubbard Brook Experimental Forest, New Hampshire, USA.

Sánchez-Salguero, R., J. J. Camarero, M. Dobbertin, Á. Fernández-Cancio, A. Vilà-Cabrera, R. D. Manzanedo, M. A. Zavala, and R. M. Navarro-Cerrillo. 2013. Contrasting vulnerability and resilience to drought-induced decline of densely planted vs. natural rear-edge Pinus nigra forests. Forest Ecology and Management 310:956-967.

Sohn, J. A., T. Gebhardt, C. Ammer, J. Bauhus, K.-H. Häberle, R. Matyssek, and T. E. Grams. 2013. Mitigation of drought by thinning: short-term and long-term effects on growth and physiological performance of Norway spruce (Picea abies). Forest Ecology and Management 308:188-197.
* * *

---

## Author Comment (AC1) · 6 May 2017

We have carefully revised the manuscript based on suggestions from the reviewers and editors and provide a response to the reviews comments that outline these changes. We express gratitude to the reviewers and editors for a strong review of our manuscript that has improved its quality.

*Anonymous Referee #1*

*General comments: In this manuscript, Lyu et al. showed that tree ring width has a pattern in both the growth (width) and its response to climate (majorly temperature) along with the latitude/altitude gradient. The creative way of using daily-step climate data implies the potential improvement in analyzing such type of time series data, e.g. tree ring analysis. It is interesting to compare/combine the latitude with altitude gradient, and would be very useful in understanding/predicting the change of tree growth in the future "warming" scenario. However, I think more efforts are needed in structuring and surmising of the paper. And I found a bit difficult to follow the results part.*

*Specific comments:*

**1)** *In terms of the climate data used in this paper, "local" meteorological station data was used in this paper, where temperature was adjusted using lapse rates in altitude gradients. Is this adjustment linearly correction? If so, this correction should have little effect on the correlation analysis. So what's the point to do this adjustment, if no absolute value of the temperature, e.g. GDD, is considered.*

**[Response]**: Yes, the climate data was adjusted using lapse rates produced by linear regression models (Kattel et al., 2013; Kattel et al., 2015) and the adjustments do not affect the correlation analyses between tree rings and the climate variables. The main

purpose of the temperature adjustments is to obtain a consistent and comparable index (July temperature) to describe the temperature conditions of the stands among both the altitudinal and latitudinal gradients. We chose to use the mean July temperature because it largely represents the temperature conditions during the main growing period in each of the forest stands.

To avoid potential misunderstandings, we further clarified the usage of the adjusted temperatures in the MS as follows: "For the Tibetan Plateau data set, we also calculated the mean July temperature for each plot. Specifically, the July temperatures were obtained based on the altitude difference between the plots and the weather station and monthly temperature lapse rates defined for the South Central Tibetan Plateau (used for SCTP) by Kattel *et al.* (2013) and for the South Eastern Tibetan Plateau (SETP) by Kattel *et al.* (2015)."

**2)** *Multiple climate factors control on tree growth was mentioned in the ms. And a number of (different) temperature signals were explained from the effect of precipitation or soil moisture, which is either non-altitude-corrected or not analyzed. Is it possible to add some analysis or information about this? In the current ms, the mentioned potential drought limitation need more evidence to support.*

**[Response]**: Thank you for the comment. We agree with the reviewer that more information about the altitude effect of precipitation is needed. When the altitude increases 100 meters, the precipitation was estimated to increase 14.3 mm in the south central part (SCTP in this study) while decreased 21.7 mm in the southeast of the plateau (SETP), and there was substantial variability within each region (Xu et al. 2007) We added the above information in the revised MS to help better interpret our results.

To provide more evidence of drought limitations on tree growth at the Tibetan sites, we calculated daily vapor pressure deficit (VPD) to represent atmospheric drought conditions, and subsequently correlated it with our tree ring index series over 31-day sliding windows (the same method used for temperatures

and precipitation). Given that tree growth was more weakly limited by precipitation for the latitudinal transect (Mäkinen et al., 2000), we did not calculate VPD for the Finland sites. Our results also showed that the most correlated precipitation was usually in a negative way (Fig. 4). We put the above points in the MS as follows:

"To better depict the potential drought limitations on tree growth, especially for the lower altitudes on the Tibetan Plateau (Lv and Zhang 2012), we calculated daily vapour pressure deficit (VPD) based on the vapour pressure (*V*) and relative air pressure (*RH*) records using Equation (1):

$$VPD = V \times \frac{1 - RH}{RH} \qquad\qquad \text{Equation (1)}$$

Since the absolute value changes would not affect the results of the latter correlation analyses between tree-ring width indices and climate variables, we did not adjust the precipitation and VPD for each plot on the Tibetan Plateau. Given that tree growth was more weakly limited by precipitation for the latitudinal transect (Mäkinen et al., 2000), we did not calculate VPD for the Finland sites."

The correlation results confirmed our expectations that the climatic drought had played an important role in limiting tree radial growth at the altitudinal gradients on the Tibetan Plateau, especially during the pre-growing season (January-April) and previous post-growing season (September-December of previous year) in the two lower altitudes of SCTP (Fig. 4j). The significantly positive correlations during July for the two higher altitudes in SETP (Fig. 4i), indicating potential temperature limitation on tree radial growth as for that lower VPD is usually accompanied with lower air temperature.

[Figure]

Fig. 4 Correlation coefficients between the ring-width index chronologies (separate sub-figures for each region, and 15 separate colors for each plot) and detrended series of mean temperature and precipitation sum in moving time-windows of 31 days. The X-axis is the central day of the time-window used in calculating the climate variables, with the first day of each month marked by ticks. The plot numbers are consistent with Table 1. Note that the color gradient legend for subpanels a-b is by 2 sites.

**3)** *Both the climate signal and the timing for the max signal are changing along with the two gradients. And in the ms, latitude and altitude gradients are both using (July) temperature to distinguish, which implies the changing temperature is one of the major reason for the change of these pattern. If that's the case, the increase of temperature during the record period should also have influence. Does this*

*mean either the timing or the strength of the temperature signal would change*
*with time? Would the warming have any effect on this analysis? Would this*
*weaken the application of using daily climate data? Because it could be always*
*changing year-by-year.*

**[Response]**: Very good comment! Temperature is indeed an important factor affecting the phenology such as the timing of budburst in spring, and may affect tree growth subsequently. According to infield observations of temperatures at a treeline altitude (4390 m a.s.l.) on the southeastern Tibetan Plateau, the onset of the growing season had advanced by 6.6 days over the period 1960-2010 (Liu et al., 2012). Similarly, spring phenology had advanced by 3-4 days per century for Scots pine (Salminen & Jalkanen 2015), and by 3-11 days per century for eight boreal tree species (Linkosalo et al. 2009). The changes in the spring phenology in this study is thus likely to be far less than 10 days given that our study periods are 60 and 39 years for Finland and Tibetan Plateau, respectively.

We agree that the inter-annual weather variations cut back the detected climate signals especially if we use a small time-window (e.g. less than 10 days). However, the used window length, compared with potential changes in the spring phenology (being likely to be far less than 10 days), is rather long (31 days), and would significantly alleviate the effects from the potential changes in the spring phenology. On the other hand, an over-long time-window is not recommended either, because it would fail to detect the most critical period of radial growth. So, the sliding window of 31-day length in this study is a result of the tradeoff between stiffness (long-term time window) and noisiness (shorter time window). We added a paragraph in the end of the Discussion Section to demonstrate the choice of window length when daily climate data is used in future studies:

"In the practice of the using daily-resolution climate data to diagnose the growth-climate relationships, the sliding window length (31 days in this study) should be a result of the tradeoff between stiffness (long-term time window) and noisiness (shorter time window). If a small time-window (e.g. less than 10 days) was used, potential noises (*i.e.* annual variations of tree phenology) might be introduced due to

Compared with commonly used monthly metrics, the application of daily weather data should thus be advantageous in revealing more accurate information about the climate signals without introducing more noises caused by inter-annual variations of the weather conditions if a sliding time window of suitable length is used.

**Reference:**

Liu, B., Y. Li, D. Eckstein, L. Zhu, B. Dawadi, and E. Liang. Has an extending growing season any effect on the radial growth of Smith fir at the timberline on the southeastern Tibetan Plateau? Trees **27**:441-446, 2013.

Linkosalo, T., Häkkinen, R., Terhivuo, J., Tuomenvirta, H., and P. Hari. The time series of flowering and leaf bud burst of boreal trees (1846–2005) support the direct temperature observations of climatic warming. Agricultural and Forest Meteorology 149, 453-461. 2009

Salminen, H., and R. Jalkanen. Modeling of bud break of Scots pine in northern Finland in 1908-2014. Frontiers in Plant Science 6, 104. 2015.

4) *There are lots of places mentioned "growing season", e.g. in the abstract, page 5 line 25, page line4 and line17, page 9 line3. However, there is no clear definition about it. If we treat the growing season as May-Aug, which was mentioned in page8 line4, it has both very strong negative and positive correlation between ring width and temperature. However, only positive correlation was mentioned in the ms. Definition and what value was used need to be cleared.*

**[Response]**: We are grateful to the reviewer for pointing that out. We added a definition of the "growing season" to the Method section as follows: "To illustrate and emphasize potential regional differences in climatic signal among the seasons, we defined four seasons (previous growing season: previous year May to August; previous post-growing season: previous year September to December; current pre-growing season: growth year January to April: current growing season: growth year May to August) and pick out the most correlated periods within each season

separately to get a more general picture of the growth-climate relationships."

We agree with the reviewer that both the positive and negative correlations should be addressed accordingly. In the revised MS, we added more descriptions on the negative correlations during the growing season as follows: "On the Tibetan Plateau, these negative correlations were accompanied with positive correlations with precipitation (Figs 4g and 5h) and the negative correlations with VPD (Fig. 4i and 4j) in the early part of the growing season, especially for the two lower altitude plots of SCTP.".

5) *Results, especially in page 5, are difficult to follow. It is hard to get consistent described results via reading the figures (Fig. 5 and 6). And the climate signals of "growing season"/"summer" made it more confusing. Does this mean a period of several months, or only one month/peak?*

**[Response]**: Thanks for pointing out this ambiguous point in the text. We wanted to find the peaks (not the averages) of correlations over 31-day windows within each predefined season (such as growing season: May-August) and compare these peaks along the altitudinal and latitudinal gradients. We replaced the "summer" by "growing season" throughout the text.

To avoid potential misunderstandings, we also made the clarification in the Method Section as follows:" To illustrate and emphasize potential regional differences in climatic signal among the seasons, we defined four seasons (previous growing season: previous year May to August; previous post-growing season: previous year September to December; current pre-growing season: growth year January to April: current growing season: growth year May to August) and pick out the most correlated periods within each season separately to get a more general picture of the growth-climate relationships."

*Other comments:*

1) *Page 3 line 27: Why the special sampling method was applied in these altitude gradient sites.*

**[Response]**: To obtain a well-replicated tree ring width chronology, at least 30 trees were sampled from each stand. Enough trees could be easily sampled from a smaller area in the middle and lower altitude stands due to larger stand densities, compared with the treeline stands. A single rectangular plot (like at treeline) with a relatively smaller areal coverage might have some exceptional conditions, such as microclimate or landform. Therefore, we sampled six groups of trees at the same altitude to avoid potential exceptional conditions for the two lower altitudes. We added these explanations in the MS as follows:

"For each of the other two altitudinal plots, a single rectangular plot covering a relatively small area (due to larger stand densities) might have some exceptional conditions, such as microclimate or landform. To increase the representativeness of the sample, six groups of Smith firs were randomly sampled. Each group was composed of a central tree and four nearest trees in different compass directions around the central tree."

2) *Page 4 line 16: the altitude of these meteorological stations should be supplied, especially for the altitude gradient sites. This would be helpful for understanding the process of altitude adjustment, and could be helpful to evaluate the consistence between the local precipitation and met record.*

**[Response]**: Thanks for the comment. The coordinates and altitudes of the stations were added as follows:" We used the Sodankylä (E 26°37', N 67°21', 179 m a.s.l.), Jyväskylä (N 62°24', E 25°40', 139 m a.s.l.) and Heinola (N 61°12', E 26°3', 92 m a.s.l.) weather stations from the northern, central and southern parts of Finland and Nielamu (N 28°11', E 85°58', 3810 m a.s.l.) and Linzhi (N 29°40', E 94°20', 2992 m a.s.l.) weather stations from the SCTP and the SETP data, respectively".

3) *Page 5 line 2: what is the difference between plot-wise mean increment chronologies and RWI.*

**[Response]**: Actually, they are the same thing. Sorry for the confusions caused by this. We replaced "plot-wise mean increment chronologies" by "RWIs".

**4)** *Not sure whether Fig. 4 is necessary. The results show more location/site dependent, which is more obvious that these gradients. And this makes sense, because RWI of similar locations should have similar pattern. And in fact, no composite cluster result is used in the following analysis.*

**[Response]**: Yes, the cluster results were not used in the subsequent analysis, so we removed the Fig. 4 and all the related descriptions and discussion in the revised MS.

**5)** *More caption is needed to describe what is the gray horizontal line in fig.5. If this is the significance test level, why all of the site has the same test level, considering the different length of climate record for different sites.*

**[Response]**: Thanks for the comment. Actually, the correlations have different test levels between Finland ($N = 60$) and Tibetan ($N = 39$) regions. We added more descriptions on the correlation periods: "The correlation periods for the Finland and the Tibetan subsets are 1938-1997 and 1968-2006, respectively. The gray horizontal lines denote the 95% significance levels."

**Anonymous Referee #2**

*The manuscript, "Tree growth and its climate signal along latitudinal and altitudinal gradients: comparison of tree rings between Finland and Tibetan Plateau" has good potential to reveal some insightful ways in which trees and tree populations respond to climate over gradients. I enjoy the potential in the data and some of the findings. It is written well enough for most audiences; some places could use some clarification. While many of the results generally follow prior work, the examination of the sub-monthly response is particularly novel, especially at such large scales.*

1) *One major concern I have is how do the authors control for differences in sunlight or day length in comparing to two regions? There are significant differences between the two and can affect the results. How do they ensure that some of the differences they see, perhaps with the response to precipitation, is not related to day length?*

**Response:** Yes, there is a significant difference in sunlight and day length between our study regions. Within the latitudinal gradient, the day length gradually changes towards higher latitudes, while no trends exist in the altitudinal transect. Increment onset indicates a tree status driven by the past winter chilling, photoperiod and thermal forcing. Development of tree status integrates signals over longer periods, but the actual onset of height and radial increment usually depends on thermal thresholds. The previous Finnish studies suggested a thermal threshold around 100 d.d. for the actual onset of radial increment. In a recent study, we found that the onset of tracheid production in Scots pine and Norway spruce varied from late May in southern Finland to mid-June in northern Finland (Jyske et al. 2014). However, no latitudinal trend was found in the temperature accumulated by the onset of tracheid production.

In the northern regions, the stimulus for increment cessation is considered to be controlled by photoperiod (e.g., Tranquillini and Unterholzner 1968; Allona et al. 2008). The variation among the years in the cessation of tracheid production indicates that other factors in addition to photoperiod determine the cessation date (e.g., Rathgeber et al. 2011; Kalliokoski et al. 2013). Recently, some studies (e.g., Tanino et

al. 2010) have shown that temperature may mediate this photoperiod response. Thus, while the difference in day length could not be avoided in the comparisons of tree growth-climate relationships between the two regions, we believe that it is one minor underlying reason for the response differences between the studied regions.

**Reference:**

Allona I, Ramnos A, Ibanez C, Contreras A, Casado R, Aragoncillo C. Molecular control of dormancy establishment in trees. Span J Agric Res 6:201–210, 2008.

Jyske T, Mäkinen H, Kalliokoski T, Nöjd P. Intra-annual xylem formation of Norway spruce and Scots pine across latitudinal gradient in Finland. Agric For Meteorol 194:241–254, 2014.

Kalliokoski T, Mäkinen H, Jyske T, Nöjd P, Linder S. Effects of nutrient optimisation on intra-annual wood formation in Norway spruce. Tree Physiol 33:1145–1155, 2013.

Rathgeber CBK, Rossi S, Bontemps J-D. Cambial activity related to tree size in a mature silver-fir plantation. Ann Bot 108:429–438, 2011.

Tanino KK, Kalcsits L, Silim S, Kendall E, Gray GR. Temperature-driven plasticity in growth cessation and dormancy development in deciduous woody plants: a working hypothesis suggesting how molecular and cellular function is affected by temperature during dormancy induction. Plant Mol Biol 73:49–65, 2010.

Tranquillini W, Unterholzner R. Dürreresistenz und Anpflanzungserfolg von Junglärchen verschiedenen Entwicklunszustandes. Centralblatt für das gesamte Forstwesen 85(2):97–110, 1968.

2) *The second smaller concern is that because the authors go to sub-monthly climate responses, Figures 5 or 6 (and much of the discussion related to those figures) are almost unnecessary. The findings are not too unexpected, but the sub-monthly analysis gives more insight. If choosing one, Figure 5 might be preferable over Figure 6. It is easier to read and provides more information.*

**Response:** We agree with the reviewer that the information of Fig. 6 is embedded in Fig. 5. However, Fig. 6 gives a more straightforward depiction of the most correlated periods along the gradients and a better comparability to the monthly correlation results (original Fig. 8). So we retained Fig. 6 in the manuscript.

*Suggestion: as this is an international paper, perhaps use only Latin names throughout the manuscript.*

**Response:** Thanks for the suggestion. We replaced the English names by Latin names through the MS. Specifically, we replaced "Norway spruce", by "*Picea abies* (L.) Karst.)", and kept the Latin names "*Abies spectabilis* (D. Don) Spach" and "*Abies georgei* var. *smithii* Viguie and Gaussen" in the MS.

*There are some concerns with the current status of the work. They are detailed below.*

**1)** *Page 2, lines 1-2: it is likely better to emphasize these are potential natural laboratories. Space for time doesn't really equal time, especially given that less than 100 years are analyzed here. Environmental variability increases with greater periods of time. So removing that from the introduction would be ok. Along these lines, the authors do not fully come back to that concept.*

   **Response:** We added "potential" before the "natural laboratories", and removed "differences, they can serve as natural laboratories to infer forest responses to global warming, using the concept of space-for-time substitution" from the MS. Thanks for pointing out this issue.

**2)** *Page 2, Line 7: perhaps "or" instead of "and" between reproduction and survival. Also, here and throughout: the Oxford comma will aid clarity in the manuscript.*

**Response:** We replaced "and" by "or" and added a comma before it in the text. We also checked the whole text and added a comma in similar occasions. Thanks for this comment.

**3)** *Page 2, Lines 8-9: There are updates to the Loehle reference in the region with greater replication at a larger spatial scale. It also indicates the same concept in this sentence. Related: explain here how a negative correlation to temperature and a positive response to precipitation together equals drought.*

**Response:** We cited the latest research (Loehle 2016) in the MS.

Higher temperature would increase the evapotranspiration and thus tend to increase drought stress on tree growth in regions with annual or seasonal water deficits (Fan et al., 2013). So, tree growth is usually negatively correlated with temperatures and positively correlated with precipitation in such environments. We added this point in the revised MS.

**Reference:**

Loehle, C., C. Idso, and T. Bently Wigley. 2016. Physiological and ecological factors influencing recent trends in United States forest health responses to climate change. Forest Ecology and Management **363**:179-189.

Fan, Z.-X., and A. Thomas. 2013. Spatiotemporal variability of reference evapotranspiration and its contributing climatic factors in Yunnan Province, SW China, 1961–2004. Climatic Change **116**:309-325.

**4)** *Page 2, Lines 26-27: (Kim and Siccama 1986) is a great forerunner of this idea and deserves recognition. It was well ahead of its time.*

**Response:** We have added the reference "Kim and Siccama 1987" in the MS. Thanks for pointing out this pioneer research.

**Reference:**

Kim, E., and Siccama, T.G.: The influence of temperature and soil moisture on the radial growth of northern hardwood tree species at Hubbard Brook Experimental Forest, New Hampshire, USA. In Proceedings, International Symposium on Ecological Aspects of Tree-Ring Analysis. Edited by G.C. Jacoby and J.W. Hornbeck. U.S. Dep. Energy Publ. CONF-8608144-26-37. pp. 26–37. 1987.

*Materials and Methods:*

**5)** *Page 3, Lines16-28: how might sampling in belts alter the climatic response? Does differing densities impact the observed climatic response? See, for instance, (SánchezSalguero et al. 2013, Sohn et al. 2013, Aldea et al. 2017).*

**Response:** Due to the changes in local climate conditions (such as temperatures and precipitation) along with increasing altitude, tree growth-climate relationships have been found to be altitude-dependent on the Tibetan Plateau (Lv & Zhang 2012; Liang

et al., 2009).

We agree with the reviewer that the tree growth-climate relationships could be affected by stand densities too. The neighbor trees could modify the microclimate around a tree and subsequently affect its radial growth. We thus added some discussions on stand densities in the revised paper. Thanks for the comment.

**6)** *Page 4, Line 4 etcetera: the proper convention is crossdate, crossdating, crossdated. Please use this convention.*

**Response:** We replaced "cross-dated", "cross-dating", and "cross-dated" by "crossdate", "crossdating", and "crossdated" throughout the text.

**7)** *Page 4, Lines 20-24: it is not likely a serious issue, but how might the inferred temperature with elevation impact the results?*

**Response:** The elevational adjustments will not affect the latter correlation analysis results. The reason why we corrected the temperatures is that we want to compare the correlations and tree-ring statistics along a common gradient (mean July temperature, in this study) to explore the potential patterns regarding tree growth-climate relationships.

**8)** *Page 4, Lines 26-29: although long in use, mean sensitivity is not a useful for comparing tree-ring records. See (Bunn et al. 2013). Please remove the MS analysis and comparison from the manuscript. Suggestion: an interesting comparison might be the coherence within each population over space. Because many of these samples were collected in plots or were aimed to be representative of the forest, perhaps a box plot or something similar expressing the strength of inter-series correlation would be compelling and insightful instead of MS.*

**Response:** Very good comment! We removed the mean sensitivity results and relevant descriptions. To depict the representativeness of the samples to the forest stands, we calculated and compared the mean inter-series correlations of all the sampling sites (Fig. 3c).

[Figure]

**Figure 3:** Stand level statistics of raw tree-ring widths along the latitudinal gradient in Finland (FI) and the altitudinal gradients on the Tibetan Plateau (TP). The lines are linear regression lines. The black colour marks the data for the latitudinal gradient, and the red denotes the data for the altitudinal gradients.

**9)** *Page 5, Lines 4-6: why 31 day windows? Did the authors experiment with narrower windows?*

**Response:** The sliding window of 31-day length in this study is a result of the tradeoff between stiffness (long-term time window) and noisiness (shorter time window). If we use a small time-window (e.g. less than 10 days) to diagnose the climate signals of tree growth, potential noises might be introduced due to the year-by-year variations of climate conditions. However, an over-long time-window is not recommended either, because it would fail to detect the most critical period of radial growth. A justification about the window length was added in the end of the Discussion Section as follows:

"In the practice of the using daily-resolution climate data to diagnose the growth-climate relationships, the sliding window length (31 days in this study) should be a result of the tradeoff between stiffness (long-term time window) and noisiness

(shorter time window). If a small time-window (e.g. less than 10 days) was used, potential noises (i.e. annual variations of tree phenology) might be introduced due to the year-by-year variations of climate conditions. However, an over-long time-window is not recommended either, because it would fail to detect the most critical period of radial growth. "

Actually, we have calculated correlations over the whole spectrum of window lengths from 5 days up to 180 days. But the comparisons and the presentations of these large amount of information among sampling sites along the temperature transects of this study would be a challenge and they are not shown because of the redundancy. For instance, the correlation results over 29-day window would resemble most of that of 30-day window. So, we pick out the correlation results over 31-day windows to demonstrate the strength of the usage of daily climate data over sliding windows, compared with commonly used monthly climate data.

**10)** *Page 5, Lines 8-11: given the submonthly work, it is not clear the use or need for these analyses. Why focus on seasons?*

**Response:** We wanted to illustrate and emphasize potential regional differences in climatic signal among the seasons. So we defined four seasons and picked out the most correlated periods within each season separately to get a more general picture of the growth-climate relationships.

**11)** *Page 5, Lines 16-19: suggest removing the MS results.*

**Response:** We removed these results as suggested.

**12)** *Page 5, Lines 20-24, Figure 4: Did the authors conduct cluster analysis within each region? Might the analysis on all populations force artificial grouping within each region? The analyses in Figures 7 & 8 somewhat supersede the scale of analysis in Figure 4, correct?*

Response: The separated cluster analyses for each of the regions kept the within-region structure of the previous results. However, given that the results of the

cluster analysis were not used in the latter growth-climate relationships, and the gradient patterns were well revealed in the latter figures as the reviewer also pointed out, we removed Fig. 4 from the revised MS.

**13)** *Suggestion for Figure 7 and most figures: consider choosing a consistent color scheme for Finland and the Tibetan Plateau and use it throughout the paper. In Figure 4 the TP is black, but by Figure 6, Finland is black. Maintaining the same colors for regions will make it easier on the reader.*

**Response:** In the revised MS, we used consistent colors and symbols in Figs 3, 6 and 7. The original Fig 4 was removed from the revised MS.

**14)** *Page 5, Lines 25-30, Figure 5: the authors write about growing season, but their information here is more specific. Suggest that in the Results the authors should be more specific. The authors make much of a negative correlation to temperature in early summer, but it appears this response is much stronger in February and March in northern Finland? Am I interpreting this incorrectly? If so, I apologize. If not, consider re-emphasizing these results. They do not seem as similar as suggested in the text. There appears to be negative correlations in northern Finland in May & June, but they are much weaker compared to earlier in the year.*

**Response:** We reorganized the description of the results and were more specific within the seasons.

As for the climate response in northern Finland, the negative correlations in February and March were stronger than in May and June. We put more emphasis on these strongest correlations in the Results section:" During pre-growing season of current year, significant and negative correlations between the RWIs and February temperatures were found on most Finnish plots (Figs 4, 5g). These negative correlation peaks showed a latitudinal pattern, with stronger correlations occurring on the northern plots (Fig. 5g)."

**15)** *To make Figure 5 easier to interpret, suggest putting the months on the top of the*

*top 2 plots. Also, perhaps make the symbols in Figure 5 smaller or replace with lines so a clearer interpretation can be made.*

**Response:** We replaced the symbols with smaller ones to avoid overlaps as much as possible. Meanwhile, we kept the month names downside because moving upward did not improve the readability after we had tried. But we did add ticks to the top axis to help for better interpretation of the figure. Please see the updated figure in the page 4.

**16)** *Page 6, Lines 10-14: why might the Finland plots have larger variability*

**Response:** We believe that the larger spatial variability of the temperature regimes (the July temperature ranges are 11.1-15.1 $^o$C, 12.4-14.8 $^o$C, 7.5-10.4 $^o$C for the Finland, SETP, and SCTP respectively) might be the reason why larger variability occurred in the critical timing of climate signal in Finland.

*Discussion:*

**17)** *Page 6, Line 20: "ring-widths are lower" than what?*

**Response:** We rephrased the sentence as follows: "Our results confirmed that ring-widths are lower at the cold ends of both latitudinal and altitudinal gradients".

**18)** *Page 6, Lines 21-23: remove MS discussion per above*

**Response:** Removed.

**19)** *Page 6, Lines 25-29: samples were collected in plots. Density, diversity, and their impact could presumably be investigated here instead of suggesting they might be at work in the results.*

**Response:** Yes, the comment is relevant. However, stand density and other local factors change along the gradient with temperature. Thus, it is difficult to quantitatively separate them from each other. We rephrased the sentences as follows:

"We believe that the temperature gradients might be confounded by local factors, such as drought and plant-plant interactions. These local factors could shape diversified habitats with varying limiting conditions other than temperatures. Previous

studies showed that tree growth on the Tibetan Plateau was affected by drought conditions (Liang et al., 2014) and competition from both trees and shrubs (Lyu et al., 2016b; Liang et al., 2016)".

20) *Page 7, Lines 15-19: there is a growing and now somewhat large body work finding or examining the relation between winter temperatures and tree growth. A review of this work would help contextualize the findings here. It might help signify the importance or the continuing line of evidence created by the findings in this study.*

**Response:** We added a review of the winter temperature effects on tree growth as follows: "… Low temperatures during wintertime were often reported to affect tree growth though bud damaging and frost desiccation (Hawkins, 1993). Increasing winter temperatures may mean less damage to leaves and buds and thus less limitation on the subsequent radial growth (Liang et al 2006; Fan et al., 2009)…".

**Reference:**

Hawkins, B.J. Photoperiod and night frost influence on the frost hardness of *Chamaecyparis nootkatensis*. Canadian Journal of Forest Research 23, 1408–1414, 1993.

Fan, Z., A. Bräuning, K. Cao, and S. Zhu. Growth-climate responses of high-elevation conifers in the central Hengduan Mountains, southwestern China. Forest Ecology and Management **258**:306-313, 2009.

Liang EY, Shao XM, Eckstein D, Huang L, Liu XH Topography- and species-dependent growth responses of *Sabina przewalskii* and *Picea crassifolia* to climate on the northeast Tibetan Plateau. For Ecol Manag 236:268–277, 2006

21) *Page 8, Lines 9-16: this is one of the most novel aspects of the study here and should be emphasized earlier and more prominently. Interesting findings.*

**Response:** Thanks for the comment. To stress these findings, we moved the sentence describing the strength of the usage of daily data ("For instance … but this association was not detected when using calendar months (Fig. 7d).") to the Result Section.

To better support our findings here, we also added a short review of the potential

mechanisms behind these results in the Discussion part as follows:

"… In northern Finland, the onset of growing season is approximately mid-June (Jyske et al., 2014). In southern Finland, the growing season roughly starts from late May (Henttonen et al., 2009). Our results showed that the timing of the most influential time period for tree growth seems to be bounded to the onsets of the growing seasons and gradually delayed from south to the north of the latitudinal gradient. Given that the early part of the growing season is critical for volume growth of trees (Cuny et al., 2015), early start of the growing season means early critical period of radial growth along the gradients. "

**Reference:**

Jyske, T., Mäkinen, H., Kalliokoski, T., and Nöjd, P.: Intra-annual tracheid production of Norway spruce and Scots pine across a latitudinal gradient in Finland, Agricultural and Forest Meteorology, 194: 241-254, 2014.

Henttonen, H. M., Mäkinen, H., and Nöjd, P.: Seasonal dynamics of the radial increment of Scots pine and Norway spruce in the southern and middle boreal zones in Finland, Canadian Journal of Forest Research, 39, 606-618, 2009.

Cuny et al. Wood biomass production lags stem-girth increase by over one month in coniferous forests. Nature Plants, 1: 15160. 2015.

**22)** *Page 8, Lines 21-23: why might this be? Is there literature that could help account for this finding?*

**Response:** The moisture limitation on tree growth was found for lower altitude plots in SCTP in our previous study (Lv and Zhang, 2012). Compared with the Finnish plots, the Tibetan sites might have higher evapotranspiration due to higher solar radiance (Leuschner, 2000), especially at lower altitudes due to higher temperatures. We integrated these points in the MS to better contextualize our study as follows:

"The moisture limitation on tree growth was found for lower altitude plots in SCTP in our previous study (Lv and Zhang, 2012). Compared with the Finnish plots, the Tibetan sites might have higher evapotranspiration due to higher solar radiance (Leuschner, 2000), especially at lower altitudes due to higher temperatures."

**Reference:**

Leuschner, C. Are high elevations in tropical mountains arid environments for plants? Ecology 81:1425-1436, 2000.

Lv, L.-X., and Q.-B. Zhang. Asynchronous recruitment history of *Abies spectabilis* along an altitudinal gradient in the Mt. Everest region. Journal of Plant Ecology **5**:147-156, 2012.

**23)** *Page 8, Lines 24-28: Masting comes out from nowhere in this manuscript. Do the species study mast? If so, how regularly? If there cannot be a better tie between the species studied and masting, it is suggested that this section be dropped.*

**Response:** We do not have long-term data on the masting behavior for both of the data sets and this part was thus removed from the revised MS.

**24)** *References:*

*Aldea, J., F. Bravo, A. Bravo-Oviedo, R. Ruiz-Peinado, F. Rodríguez, and M. del Río. 2017. Thinning enhances the species-specific radial increment response to drought in Mediterranean pine-oak stands. Agricultural and Forest Meteorology 237:371-383.*

*Bunn, A. G., E. Jansma, M. Korpela, R. D. Westfall, and J. Baldwin. 2013. Using simulations and data to evaluate mean sensitivity (_) as a useful statistic in dendrochronology. Dendrochronologia 31:250-254.*

*Kim, E., and T. G. Siccama. 1986. The influence of temperature and soil moisture on the radial growth of northern hardwood tree species at Hubbard Brook Experimental Forest, New Hampshire, USA.*

*Sánchez-Salguero, R., J. J. Camarero, M. Dobbertin, Á. Fernández-Cancio, A. Vilà-Cabrera, R. D. Manzanedo, M. A. Zavala, and R. M. Navarro-Cerrillo. 2013. Contrasting vulnerability and resilience to drought-induced decline of densely planted vs. natural rear-edge Pinus nigra forests. Forest Ecology and Management 310:956-967.*

*Sohn, J. A., T. Gebhardt, C. Ammer, J. Bauhus, K.-H. Häberle, R. Matyssek, and T. E. Grams. 2013. Mitigation of drought by thinning: short-term and long-term effects on growth and physiological performance of Norway spruce (Picea abies). Forest Ecology and Management 308:188-197.*

**Response:** We are very grateful to the reviewer for the insightful comments and the related references.

Yours sincerely,

Lixin Lyu     on behalf of all co-authors

---

## Author Comment (AC2) · 6 May 2017

We put all of our responses to the referees in one letter, please find it in the supplement to the "AC1".

---

## Author Response (AR1)

We have carefully revised the manuscript based on suggestions from the reviewers and editors and provide a response to the reviews comments that outline these changes. We express gratitude to the reviewers and editors for a strong review of our manuscript that has significantly improved its quality.

*Anonymous Referee #1*

***General comments***: *In this manuscript, Lyu et al. showed that tree ring width has a pattern in both the growth (width) and its response to climate (majorly temperature) along with the latitude/altitude gradient. The creative way of using daily-step climate data implies the potential improvement in analyzing such type of time series data, e.g. tree ring analysis. It is interesting to compare/combine the latitude with altitude gradient, and would be very useful in understanding/predicting the change of tree growth in the future "warming" scenario. However, I think more efforts are needed in structuring and surmising of the paper. And I found a bit difficult to follow the results part.*

***Specific comments***:

1) *In terms of the climate data used in this paper, "local" meteorological station data was used in this paper, where temperature was adjusted using lapse rates in altitude gradients. Is this adjustment linearly correction? If so, this correction should have little effect on the correlation analysis. So what's the point to do this adjustment, if no absolute value of the temperature, e.g. GDD, is considered.*

**[Response]**: Yes, the climate data was adjusted using lapse rates produced by linear regression models (Kattel et al., 2013; Kattel et al., 2015) and the adjustments do not affect the correlation analyses between tree rings and the climate variables. The main

purpose of the temperature adjustments is to obtain a consistent and comparable index (July temperature) to describe the temperature conditions of the stands among both the altitudinal and latitudinal gradients. We chose to use the mean July temperature because it largely represents the temperature conditions during the main growing period in each of the forest stands.

To avoid potential misunderstandings, we further clarified the usage of the adjusted temperatures in the MS as follows: "We also calculated the mean July temperature for each plot of the Tibetan Plateau data set. July temperatures were obtained based on the altitude differences between the plots and weather station and monthly temperature lapse rates were defined for the South Central Tibetan Plateau (used for SCTP) by Kattel *et al.* (2013) and for the South Eastern Tibetan Plateau (SETP) by Kattel *et al.* (2015)." (Page 4 Line 29-31 Page 5 Line 1)

2) *Multiple climate factors control on tree growth was mentioned in the ms. And a number of (different) temperature signals were explained from the effect of precipitation or soil moisture, which is either non-altitude-corrected or not analyzed. Is it possible to add some analysis or information about this? In the current ms, the mentioned potential drought limitation need more evidence to support.*

**[Response]**: Thank you for the comment. We agree with the reviewer that more information about the altitude effect of precipitation is needed. When the altitude increases 100 meters, the precipitation was estimated to increase 14.3 mm in the south central part (SCTP in this study) while decreased 21.7 mm in the southeast of the plateau (SETP), and there was substantial variability within each region (Lu et al. 2007) We added the above information in the revised MS to help better interpret our results.

To provide more evidence of drought limitations on tree growth at the Tibetan sites, we calculated daily vapor pressure deficit (VPD) to represent atmospheric drought conditions, and subsequently correlated it with our tree ring index series over 31-day sliding windows (the same method used for temperatures

and precipitation). Given that tree growth was more weakly limited by precipitation for the latitudinal transect (Mäkinen et al., 2000), we did not calculate VPD for the Finland sites. Our results also showed that the most correlated precipitation was usually in a negative way (Fig. 4). We put the above points in the MS as follows:

"To depict the potential drought limitations on tree growth, we calculated the daily vapour pressure deficit (VPD) based on vapour pressure (*V*) and relative air pressure (*RH*) records, using Equation (1) (Allen et al., 1998):

$$\text{VPD} = V \times \frac{1 - RH}{RH} \qquad\qquad \text{Equation (1)}$$

Since absolute-value changes would not affect the results of the correlation analyses between tree-ring width indices and climate variables, the precipitation and VPD were not adjusted for each plot on the Tibetan Plateau. Given that tree growth was only weakly limited by precipitation on the latitudinal gradient (Mäkinen et al., 2000), VPD was not calculated for the Finnish sites." (Page 5 Line 5-11)

The correlation results confirmed our expectations that the climatic drought had played an important role in limiting tree radial growth at the altitudinal gradients on the Tibetan Plateau, especially during the pre-growing season (January-April) and previous post-growing season (September-December of previous year) in the two lower altitudes of SCTP (Fig. 4j). The significantly positive correlations during July for the two higher altitudes in SETP (Fig. 4i), indicating potential temperature limitation on tree radial growth as for that lower VPD is usually accompanied with lower air temperature.

[Figure]

Fig. 4 Correlation coefficients between the ring-width index chronologies (separate sub-figures for each region, and 15 separate colors for each plot) and detrended series of mean temperature and precipitation sum in moving time-windows of 31 days. The X-axis is the central day of the time-window used in calculating the climate variables, with the first day of each month marked by ticks. The plot numbers are consistent with Table 1. Note that the color gradient legend for subpanels a-b is by 2 sites.

**3)** *Both the climate signal and the timing for the max signal are changing along with the two gradients. And in the ms, latitude and altitude gradients are both using (July) temperature to distinguish, which implies the changing temperature is one of the major reason for the change of these pattern. If that's the case, the increase of temperature during the record period should also have influence. Does this*

*mean either the timing or the strength of the temperature signal would change with time? Would the warming have any effect on this analysis? Would this weaken the application of using daily climate data? Because it could be always changing year-by-year.*

**[Response]**: Very good comment! Temperature is indeed an important factor affecting the phenology such as the timing of budburst in spring, and may affect tree growth subsequently. According to infield observations of temperatures at a treeline altitude (4390 m a.s.l.) on the southeastern Tibetan Plateau, the onset of the growing season had advanced by 6.6 days over the period 1960-2010 (Liu et al., 2012). Similarly, spring phenology had advanced by 3-4 days per century for Scots pine (Salminen & Jalkanen 2015), and by 3-11 days per century for eight boreal tree species (Linkosalo et al. 2009). The changes in the spring phenology in this study is thus likely to be far less than 10 days given that our study periods are 60 and 39 years for Finland and Tibetan Plateau, respectively.

We agree that the inter-annual weather variations cut back the detected climate signals especially if we use a small time-window (e.g. less than 10 days). However, the used window length, compared with potential changes in the spring phenology (being likely to be far less than 10 days), is rather long (31 days), and would significantly alleviate the effects from the potential changes in the spring phenology. On the other hand, an over-long time-window is not recommended either, because it would fail to detect the most critical period of radial growth. So, the sliding window of 31-day length in this study is a result of the tradeoff between stiffness (long-term time window) and noisiness (shorter time window). We added a paragraph in the end of the Discussion Section to demonstrate the choice of window length when daily climate data is used in future studies:

"The sliding window length of 31 days used in this study was a result of a trade-off between stiffness (longer time window) and noisiness (shorter time window). If a small time-window (e.g., below 10 days) was used, noisy correlation patterns were introduced due to year-by-year variations of climate conditions. However, an overly long time-window is not recommended either, as it fails to detect the most

critical period of radial growth. " (Page 9 Line 15-18)

Compared with commonly used monthly metrics, the application of daily weather data should thus be advantageous in revealing more accurate information about the climate signals without introducing more noises caused by inter-annual variations of the weather conditions if a sliding time window of suitable length is used.

**Reference:**

Liu, B., Y. Li, D. Eckstein, L. Zhu, B. Dawadi, and E. Liang. Has an extending growing season any effect on the radial growth of Smith fir at the timberline on the southeastern Tibetan Plateau? Trees **27**:441-446, 2013.

Linkosalo, T., Häkkinen, R., Terhivuo, J., Tuomenvirta, H., and P. Hari. The time series of flowering and leaf bud burst of boreal trees (1846–2005) support the direct temperature observations of climatic warming. Agricultural and Forest Meteorology 149, 453-461. 2009

Salminen, H., and R. Jalkanen. Modeling of bud break of Scots pine in northern Finland in 1908-2014. Frontiers in Plant Science 6, 104. 2015.

4) *There are lots of places mentioned "growing season", e.g. in the abstract, page 5 line 25, page line4 and line17, page 9 line3. However, there is no clear definition about it. If we treat the growing season as May-Aug, which was mentioned in page8 line4, it has both very strong negative and positive correlation between ring width and temperature. However, only positive correlation was mentioned in the ms. Definition and what value was used need to be cleared.*

**[Response]**: We are grateful to the reviewer for pointing that out. We added a definition of the "growing season" to the Method section as follows: "To illustrate and emphasize potential regional differences in climatic signals among the seasons, we defined four seasons (previous growing season: May to August of the previous year; previous post-growing season: September to December of the previous year; current pre-growing season: January to April of the growth year; current growing season: May to August of the growth year) and separately chose the most correlated periods within each season to arrive at a more general picture of the growth-climate relationships." (Page 5 Line 24-28)

We agree with the reviewer that both the positive and negative correlations should be addressed accordingly. In the revised MS, we added more descriptions on the negative correlations during the growing season as follows: "On the Tibetan Plateau, these negative correlations were accompanied by positive correlations with precipitation (Figs. 4g and 5h) and negative correlations with VPD (Fig. 4i and 4j) during the early part of the growing season, especially for the two lower altitude plots of SCTP.". (Page 6 Line 10-13)

5) *Results, especially in page 5, are difficult to follow. It is hard to get consistent described results via reading the figures (Fig. 5 and 6). And the climate signals of "growing season"/"summer" made it more confusing. Does this mean a period of several months, or only one month/peak?*

**[Response]**: Thanks for pointing out this ambiguous point in the text. We wanted to find the peaks (not the averages) of correlations over 31-day windows within each predefined season (such as growing season: May-August) and compare these peaks along the altitudinal and latitudinal gradients. We replaced the "summer" by "growing season" throughout the text.

To avoid potential misunderstandings, we also made the clarification in the Method Section as follows:"To illustrate and emphasize potential regional differences in climatic signals among the seasons, we defined four seasons (previous growing season: May to August of the previous year; previous post-growing season: September to December of the previous year; current pre-growing season: January to April of the growth year; current growing season: May to August of the growth year) and separately chose the most correlated periods within each season to arrive at a more general picture of the growth-climate relationships." (Page 5 Line 24-28)

*Other comments*:

1) *Page 3 line 27: Why the special sampling method was applied in these altitude gradient sites.*

**[Response]**: To obtain a well-replicated tree ring width chronology, at least 30 trees

were sampled from each stand. Enough trees could be easily sampled from a smaller area in the middle and lower altitude stands due to larger stand densities, compared with the treeline stands. A single rectangular plot (like at treeline) with a relatively smaller areal coverage might have some exceptional conditions, such as microclimate or landform. Therefore, we sampled six groups of trees at the same altitude to avoid potential exceptional conditions for the two lower altitudes. We added these explanations in the MS as follows:

"For each of the remaining two altitudinal plots, using a single rectangular plot over a relatively small areal range (due to larger stand densities) might reflect very specific conditions, such as particular microclimates and landforms of the plot. To increase the representativeness of forest stands, six random groups of *A. georgei* were sampled. Each group consisted of a central tree and four nearest trees in different compass directions around the central tree." (Page 4 Line 4-7)

2) *Page 4 line 16: the altitude of these meteorological stations should be supplied, especially for the altitude gradient sites. This would be helpful for understanding the process of altitude adjustment, and could be helpful to evaluate the consistence between the local precipitation and met record.*

[Response]: Thanks for the comment. The coordinates and altitudes of the stations were added as follows:"We used the Sodankylä (E 26°37', N 67°21', 179 m a.s.l.), Jyväskylä (N 62°24', E 25°40', 139 m a.s.l.), and Heinola (N 61°12', E 26°3', 92 m a.s.l.) weather stations from the northern, central, and southern parts of Finland and the weather stations Nielamu (N 28°11', E 85°58', 3810 m a.s.l.) and Linzhi (N 29°40', E 94°20', 2992 m a.s.l.) from the SCTP and the SETP data, respectively (Fig. 2)". (Page 4 Line 22-25)

3) *Page 5 line 2: what is the difference between plot-wise mean increment chronologies and RWI.*

[Response]: Actually, they are the same thing. Sorry for the confusions caused by this. We replaced "plot-wise mean increment chronologies" by "RWIs". (Page 5 Line 18)

**4)** *Not sure whether Fig. 4 is necessary. The results show more location/site dependent, which is more obvious that these gradients. And this makes sense, because RWI of similar locations should have similar pattern. And in fact, no composite cluster result is used in the following analysis.*

**[Response]**: Yes, the cluster results were not used in the subsequent analysis, so we removed the Fig. 4 and all the related descriptions and discussion in the revised MS.

**5)** *More caption is needed to describe what is the gray horizontal line in fig.5. If this is the significance test level, why all of the site has the same test level, considering the different length of climate record for different sites.*

**[Response]**: Thanks for the comment. Actually, the correlations have different test levels between Finland ($N = 60$) and Tibetan ($N = 39$) regions. We added more descriptions on the correlation periods: "The correlation periods for the Finnish and the Tibetan subsets are 1938-1997 and 1968-2006, respectively. The gray horizontal lines denote the 5% significance levels." (Page 17 Line 15-16)

**Anonymous Referee #2**

*The manuscript, "Tree growth and its climate signal along latitudinal and altitudinal gradients: comparison of tree rings between Finland and Tibetan Plateau" has good potential to reveal some insightful ways in which trees and tree populations respond to climate over gradients. I enjoy the potential in the data and some of the findings. It is written well enough for most audiences; some places could use some clarification. While many of the results generally follow prior work, the examination of the sub-monthly response is particularly novel, especially at such large scales.*

**1)** *One major concern I have is how do the authors control for differences in sunlight or day length in comparing to two regions? There are significant differences between the two and can affect the results. How do they ensure that some of the differences they see, perhaps with the response to precipitation, is not related to day length?*

**Response:** Yes, there is a significant difference in sunlight and day length between our study regions. Within the latitudinal gradient, the day length gradually changes towards higher latitudes, while no trends exist in the altitudinal transect. Increment onset indicates a tree status driven by the past winter chilling, photoperiod and thermal forcing. Development of tree status integrates signals over longer periods, but the actual onset of height and radial increment usually depends on thermal thresholds. The previous Finnish studies suggested a thermal threshold around 100 d.d. for the actual onset of radial increment. In a recent study, we found that the onset of tracheid production in Scots pine and Norway spruce varied from late May in southern Finland to mid-June in northern Finland (Jyske et al. 2014). However, no latitudinal trend was found in the temperature accumulated by the onset of tracheid production.

In the northern regions, the stimulus for increment cessation is considered to be controlled by photoperiod (e.g., Tranquillini and Unterholzner 1968; Allona et al. 2008). The variation among the years in the cessation of tracheid production indicates that other factors in addition to photoperiod determine the cessation date (e.g., Rathgeber et al. 2011; Kalliokoski et al. 2013). Recently, some studies (e.g., Tanino et

al. 2010) have shown that temperature may mediate this photoperiod response. Thus, while the difference in day length could not be avoided in the comparisons of tree growth-climate relationships between the two regions, we believe that it is one minor underlying reason for the response differences between the studied regions.

*many of these samples were collected in plots or were aimed to be representative of the forest, perhaps a box plot or something similar expressing the strength of inter-series correlation would be compelling and insightful instead of MS.*

   **Response:** Very good comment! We removed the mean sensitivity results and relevant descriptions. To depict the representativeness of the samples to the forest stands, we calculated and compared the mean inter-series correlations of all the sampling sites (Fig. 3c).

[Figure]

**Figure 3:** Stand level statistics of raw tree-ring widths along the latitudinal gradient in Finland (FI) and the altitudinal gradients on the Tibetan Plateau (TP). The lines are linear regression lines. The black colour marks the data for the latitudinal gradient, and the red denotes the data for the altitudinal gradients.

**9)** *Page 5, Lines 4-6: why 31 day windows? Did the authors experiment with narrower windows?*

**Response:** The sliding window of 31-day length in this study is a result of the tradeoff between stiffness (long-term time window) and noisiness (shorter time window). If we use a small time-window (e.g. less than 10 days) to diagnose the climate signals of

tree growth, potential noises might be introduced due to the year-by-year variations of climate conditions. However, an over-long time-window is not recommended either, because it would fail to detect the most critical period of radial growth. A justification about the window length was added in the end of the Discussion Section as follows:

"The sliding window length of 31 days used in this study was a result of a trade-off between stiffness (longer time window) and noisiness (shorter time window). If a small time-window (e.g., below 10 days) was used, noisy correlation patterns were introduced due to year-by-year variations of climate conditions. However, an overly long time-window is not recommended either, as it fails to detect the most critical period of radial growth. " (Page 9 Line 15-18)

Actually, we have calculated correlations over the whole spectrum of window lengths from 5 days up to 180 days. But the comparisons and the presentations of these large amount of information among sampling sites along the temperature transects of this study would be a challenge and they are not shown because of the redundancy. For instance, the correlation results over 29-day window would resemble most of that of 30-day window. So, we pick out the correlation results over 31-day windows to demonstrate the strength of the usage of daily climate data over sliding windows, compared with commonly used monthly climate data.

**10)** *Page 5, Lines 8-11: given the submonthly work, it is not clear the use or need for these analyses. Why focus on seasons?*

**Response:** We wanted to illustrate and emphasize potential regional differences in climatic signal among the seasons. So we defined four seasons and picked out the most correlated periods within each season separately to get a more general picture of the growth-climate relationships. (Page 5 Line 24-28)

**11)** *Page 5, Lines 16-19: suggest removing the MS results.*

**Response:** We removed these results regarding MS as suggested.

**12)** *Page 5, Lines 20-24, Figure 4: Did the authors conduct cluster analysis within*

*each region? Might the analysis on all populations force artificial grouping within each region? The analyses in Figures 7 & 8 somewhat supersede the scale of analysis in Figure 4, correct?*

Response: The separated cluster analyses for each of the regions kept the within-region structure of the previous results. However, given that the results of the cluster analysis were not used in the latter growth-climate relationships, and the gradient patterns were well revealed in the latter figures as the reviewer also pointed out, we removed Fig. 4 from the revised MS.

**13)** *Suggestion for Figure 7 and most figures: consider choosing a consistent color scheme for Finland and the Tibetan Plateau and use it throughout the paper. In Figure 4 the TP is black, but by Figure 6, Finland is black. Maintaining the same colors for regions will make it easier on the reader.*

**Response:** In the revised MS, we used consistent colors and symbols in Figs 3, 6 and 7. The original Fig 4 was removed from the revised MS.

**14)** *Page 5, Lines 25-30, Figure 5: the authors write about growing season, but their information here is more specific. Suggest that in the Results the authors should be more specific. The authors make much of a negative correlation to temperature in early summer, but it appears this response is much stronger in February and March in northern Finland? Am I interpreting this incorrectly? If so, I apologize. If not, consider re-emphasizing these results. They do not seem as similar as suggested in the text. There appears to be negative correlations in northern Finland in May & June, but they are much weaker compared to earlier in the year.*

**Response:** We reorganized the description of the results and were more specific within the seasons.

As for the climate response in northern Finland, the negative correlations in February and March were stronger than in May and June. We put more emphasis on these strongest correlations in the Results section:"During the current pre-growing season, significant negative correlations between the RWIs and February temperatures

were found for most Finnish plots (Figs. 4, 5g). These negative correlation peaks showed a latitudinal pattern, with stronger correlations for northern plots (Fig. 5g)." (Page 6 Line 14-16)

**15)** *To make Figure 5 easier to interpret, suggest putting the months on the top of the top 2 plots. Also, perhaps make the symbols in Figure 5 smaller or replace with lines so a clearer interpretation can be made.*

**Response:** We replaced the symbols with smaller ones to avoid overlaps as much as possible. Meanwhile, we kept the month names downside because moving upward did not improve the readability after we had tried. But we did add ticks to the top axis to help for better interpretation of the figure. Please see the updated figure in the page 4.

**16)** *Page 6, Lines 10-14: why might the Finland plots have larger variability*

**Response:** We believe that the larger spatial variability of the temperature regimes (the July temperature ranges are 11.1-15.1 $^{\circ}$C, 12.4-14.8 $^{\circ}$C, 7.5-10.4 $^{\circ}$C for the Finland, SETP, and SCTP respectively) might be the reason why larger variability occurred in the critical timing of climate signal in Finland.

*Discussion:*

**17)** *Page 6, Line 20: "ring-widths are lower" than what?*

**Response:** We rephrased the sentence as follows: "Our results confirmed that ring-widths are lower at the cold ends of both latitudinal and altitudinal gradients than at the warm ends". (Page 7 Line 8-9)

**18)** *Page 6, Lines 21-23: remove MS discussion per above*

**Response:** Removed.

**19)** *Page 6, Lines 25-29: samples were collected in plots. Density, diversity, and their impact could presumably be investigated here instead of suggesting they might be at work in the results.*

**Response:** Yes, the comment is relevant. However, stand density and other local factors change along the gradient with temperature. Thus, it is difficult to quantitatively separate them from each other. We rephrased the sentences as follows:

"We suggest that the temperature gradients might be influenced by local factors, such as drought and plant-plant interactions. Local factors such as stand density and landform could shape diversified habitats with varying limiting conditions beyond temperatures. Previous studies showed tree growth on the Tibetan Plateau to be affected by drought conditions (Liang et al., 2014) as well as by competition from both trees and shrubs (Lyu et al., 2016b; Liang et al., 2016)". (Page 7 Line 15-18)

20) *Page 7, Lines 15-19: there is a growing and now somewhat large body work finding or examining the relation between winter temperatures and tree growth. A review of this work would help contextualize the findings here. It might help signify the importance or the continuing line of evidence created by the findings in this study.*

**Response:** We added a review of the winter temperature effects on tree growth as follows: "…Low temperatures were often reported to affect tree growth through bud damaging and frost desiccation (Hawkins, 1993). Increasing temperatures may cause less damage to leaves and buds and thus, be less limiting for subsequent radial growth (Liang et al 2006; Fan et al., 2009)…". (Page 8 Line 6-8)

**Response:** We are very grateful to the reviewer for the insightful comments and the related references.

In addition to these above mentioned changes, we also sent our MS to a British editor to improve the English language of the text through a professional language service agency (MogoEdit, http://en.mogoedit.com/) and attached the certificate of English Editing at the end of this response letter. Note that the mark-up MS was attached below the response letter.

Yours sincerely,

Lixin Lyu     on behalf of all co-authors

[revised manuscript text omitted]

[Figure]

[Figure]

**Figure 6**

[Figure]

**Figure 7**

[Figure]

[Figure]

**CERTIFICATE OF ENGLISH EDITING**

This is to certify that the manuscript entitled

***Tree growth and its climate signal along latitudinal and altitudinal gradients: comparison of tree rings between Finland and Tibetan Plateau***

commissioned to us has been carefully edited by a native English-speaking editor of MogoEdit, and the grammar, spelling, and punctuation have been verified and corrected where needed. Based on this review, we believe that the language in this paper meets academic journal requirements. Please contact us with any questions.

[Figure]

*Gang Zhang*

Dr. Gang Zhang
Founder & CEO of MogoEdit

Date of Issue
May 14, 2017

**Disclaimer:** The changes in the document may be accepted or rejected by the authors in their sole discretion after our editing. However, MogoEdit is not responsible for revisions made to the document after our edit on **May 14, 2017**.

MogoEdit is a professional English editing company who provides English language editing, translation, and publication support services to individuals and corporate customers worldwide. As a company invested by the affiliate fund of Chinese Academy of Science, MogoEdit is one of the leading language editing service providers in China, whose clients come from more than 1000 universities and research institutes.

MogoEdit Website:   http://en.mogoedit.com/

500+ native English editors:   http://en.mogoedit.com/editors

[Figure]

Mogo Internet Technology Co., LTD.

No. 25, 1st Gaoxin Road, Xi'an 710075, PR China +86 02988317483 support@mogoedit.com